# AttackSeqBench: Benchmarking Large Language Models in Analyzing Attack Sequences within Cyber Threat Intelligence

## Abstract

Cyber Threat Intelligence (CTI) reports document observations of cyber threats, synthesizing evidence about adversaries' actions and intent into actionable knowledge that informs detection, response, and defense planning. However, the unstructured and verbose nature of CTI reports poses significant challenges for security practitioners to manually extract and analyze such sequences. Although large language models (LLMs) exhibit promise in cybersecurity tasks such as entity extraction and knowledge graph construction, their understanding and reasoning capabilities towards behavioral sequences remains underexplored. To address this, we introduce **AttackSeqBench**, a benchmark designed to systematically evaluate LLMs' reasoning abilities across the tactical, technical, and procedural dimensions of adversarial behaviors, while satisfying Extensibility, Reasoning Scalability, and Domain-dpecific Epistemic Expandability. We further benchmark 7 LLMs, 5 LRMs and 4 post-training strategies across the proposed 3 benchmark settings and 3 benchmark tasks within our **AttackSeqBench** to identify their advantages and limitations in such specific domain. Our findings contribute to a deeper understanding of LLM-driven CTI report understanding and foster its application in cybersecurity operations. Our code and dataset are available at: https://anonymous.4open.science/r/AttackSeqBench.

## 1 Introduction

Amid rapid digital transformation, the increasing sophistication and diversity of cyber attacks have become a pervasive concern for cybersecurity globally (Duo et al., 2022). Cyber Threat Intelligence (CTI) reports, which document observations of these threats, have emerged as a crucial resource in proactive defenses (Wagner et al., 2019). However, they are often lengthy and unstructured, resulting in a labor-intensive task for practitioners to manually analyze and extract insights (Sun et al., 2023).

Recently, Large Language Models (LLMs) have demonstrated promising potential in several cybersecurity applications (Zhang et al., 2024a). This sheds new light towards incorporating LLMs into CTI Report Understanding (CRU) task, where we define CRU as a broad concept encompassing tasks that derive and reason threat intelligence from CTI reports. However, existing benchmarks primarily assess LLMs on threat intelligence extraction and attack attribution, while their potential for understanding adversarial behaviors dependencies in CTI reports remains largely unexplored (*cf.* Appendix A.7). Such ability is crucial in anticipating future malicious attack actions, particularly in multi-stage cyber attacks launched by Advanced Persistent Threats (APTs) (Li et al., 2022).

As illustrated in Figure 1, we define the sequence of adversary behaviors as *attack sequence* (Al-Sada et al., 2025) to represent the execution flow of malicious actions across different stages of a cyber attack under the MITRE ATT&CK® framework (Strom et al., 2018). Building on this definition and the following key perspectives, we further delve into the suitability of LLMs in analyzing *attack sequences*. 1) *Extensibility*: To address the ever-evolving threat landscape and the advancements of LLMs, the proposed benchmark must be extensible to incorporate *attack sequences* from newly observed CTI reports. 2) *Reasoning Scalability*: Recently, Large Reasoning Models (LRMs) have demonstrated substantial advantages over conventional LLMs in multi-step reasoning tasks, such as coding and mathematical reasoning. However, existing CRU works have primarily focused on ad-

Figure 1: Illustration an example cyber *attack sequence* and our `AttackSeqBench`.

dressing CTI–related tasks via LLM, leaving the necessity of reasoning for *attack sequence* analysis largely unexplored. 3) ***Domain-Specific Epistemic Expandability***: LLMs exhibited limitations in factual reliability on knowledge-intensive tasks (Xu et al., 2024b), analogously, LLM-driven CRU, which requires specialized cybersecurity knowledge, is also subject to such limitations. This requirement becomes particularly pronounced in *attack sequence* analysis, which necessitates a comprehensive understanding of adversarial behaviors to effectively reason multi-stage cyber attacks.

Building upon these perspectives, we introduce `AttackSeqBench`, a novel benchmark designed for comprehensive evaluation of LLMs in *attack sequence* analysis. Catering to ***Extensibility***, we first construct *attack sequences* based on extensive real-world CTI reports, ensuring that the benchmark accurately reflects the complexity and diversity of Tactics, Techniques, and Procedures (TTPs) in cyber attacks performed by APTs. Moreover, we design three Question Answering (Q&A) tasks under the adversary behaviors hierarchy in MITRE ATT&CK® and develop an automated Q&A generation pipeline that converts newly-collected CTI reports into the pre-defined format, enabling its extensibility on the corpus side. Following ***Reasoning Scalability***, we further evaluate several LRMs and reasoning distillation strategies, which function well in general domains, to identify their strengths and limitations on the specialized *attack sequence* analysis task, providing helpful insights for future research in this area. To achieve ***Domain-Specific Epistemic Expandability***, we aggregate cybersecurity-related knowledge from some existing benchmarks and embed it into LLMs via several post-training strategies to examine their epistemic expandability at the model level. Moreover, we also extend beyond the conventional zero-shot setting by introducing context-based and RAG-empowered settings, which pertinently assess LLMs' epistemic expandability when injecting domain-specific cybersecurity knowledge at the semantic and representation levels.

Our contribution are as follows: (I) We introduce `AttackSeqBench`, the pioneering benchmark that systematically evaluates the ability of existing LLMs, LRMs, and post-training strategies to analyze *attack sequences* across diverse inference settings and multi-level tasks. (*cf.* Section 2) (II) We quantitatively demonstrate that existing LRMs fail to substantially outperform LLMs on *attack sequence* analysis and perform markedly worse in most cases, a contrast to their advantages observed in domains such as mathematics and coding. (*cf.* Section 3.3) (III) We offer a comprehensive analysis of how parameterization and parameter scale affect existing models' *attack sequence* analysis, and further examine why current LRMs and RAG underperform on this specialized task. This work uncovers the fundamental limitations of current models in *attack sequence* analysis and provides actionable insights to guide future research in this domain. (*cf.* Section 3.4 and Section 3.5)

## 2    DATASET CONSTRUCTION AND VERIFICATION

**Goal of our `AttackSeqBench`**

Our goal is to *explore the capability of diverse types of LLMs in attack sequence understanding*. Through comprehensive evaluation across various tasks and settings, we emphasize the strengths and limitations of existing LLMs, offering the promising yet underexplored directions.

### 2.1    PROBLEM DEFINITIONS

CTI report understanding aims to convert the unstructural report into the structural formulation and further comprehend the sequential attack patterns of the structured threat intelligence knowledge. To achieve this, we define the attack sequence $S$ as the progression of adversarial behaviors described in a given CTI report, characterized by the logical order of TTPs based on their associated tactics within the ATT&CK KB. Formally, we utilize a 4-tuple to represent $S$ as $S = (T, E, P, O)$, where:

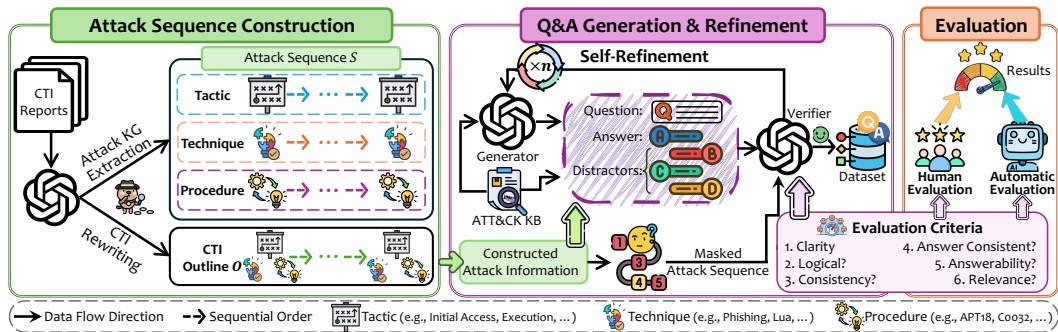

Figure 2: Overview of our automated QA dataset construction pipeline.

- *Tactic Sequence $T$*: An ordered list of ATT&CK tactics, such that $T = (t_1, \ldots, t_n)$, where $t_k$ is the $k$-th tactic in the sequence.
- *Technique Mappings $E$*: The set of ATT&CK techniques / sub-techniques in $S$, where $E(t_k) = \{e_{1,k}, \ldots, e_{i_k,k}\}$ denote all the techniques / sub-techniques that belong to tactic $t_k$.
- *Procedure Mappings $P$*: The set of ATT&CK procedures in $S$, where each procedure is represented as a triplet $p = (\texttt{subject}, \texttt{action}, \texttt{object})$. Here, we leverage $P(e_{j,k}) = \{p_{1,j,k}, \ldots, p_{m,j,k}\}$ to describe the set of procedure triplets of the technique $e_{j,k} \in E(t_k)$.
- *CTI Outline $O$*: A textual summary of the organized TTPs based on the order of *Tactic Sequence $T$*, such that $O = (o_1, \ldots, o_n)$, where $o_k$ refers to the summarized text associated with tactic $t_k$.

## 2.2 DATASET CONSTRUCTION

As illustrated in Figure 2, we first construct *attack sequences* using the extracted TTPs and CTI outline from CTI reports. Then, we generate Q&A pairs based on the constructed *attack sequences* and refine them based on a tailored evaluate criteria before populating the QA dataset.

**Attack Sequence Construction.** To efficiently and massively extract threat intelligence from these unstructured reports, we utilize a set of 408 CTI reports from various security vendors (Cisco Talos Intelligence Group, 2025; Microsoft, 2025) to construct *attack sequences* that accurately reflect the behaviors of real-world APTs. Specifically, we utilize a LLM-based KG construction framework (Zhang et al., 2025c) to automatically parse CTI reports, extract TTPs from each chunk into three level, generate CTI outlines, and combin them to construct the *attack sequences $S$*. Notably, we exclude CTI outlines which contains less than two ATT&CK tactics in *attack sequence* construction as they are unlikely to detail attack patterns observed in real-world cyber attacks.

**Q&A Generation.** Inspired by the remarkable question generation abilities of LLMs across multiple domains (Alam et al., 2024; Zhang et al., 2024b; Mucciaccia et al., 2025), we adopt an answer-aware question generation approach using GPT-4o (OpenAI, 2024a). To elaborate, we first instruct the LLM to generate a seed Q&A pair for each tactic, technique, and group of procedures with the given *attack sequence*. Furthermore, we utilize the model's In-Context Learning ability to generate the more relevant Q&A pairs (Dong et al., 2024), by including the CTI outline and few-shot Q&A examples in the question generation prompt (*cf.* Appendix C.1).

For the Multiple-Choice Question (MCQ) tasks, we adopt a rule-based approach to select three choices as distractors. Specifically, we select a adjacent tactic of tactic $t_k$ within the *Tactic Sequence $T$* (*i.e.*, $t_{k+1}$ or $t_{k-1}$) and randomly select two tactics from the ATT&CK KB in *AttackSeq-Tactic*. Regarding *AttackSeq-Technique*, we follow the STARC annotation framework (Berzak et al., 2020) to define the selection rules with the given technique $e_{j,k}$: (1) The first technique belongs to the same tactic $t_k$ but not present in the given *attack sequence*, *i.e.*, $e_{i,k} \notin E(t_k)$; (2) The second technique is supported by the given *attack sequence* but belongs to another tactic $e_{j,j} \notin E(t_k)$; (3) The third technique comes from a randomly chosen tactic that is not supported by the given *attack sequence*.

Regarding the Yes-No Question tasks, we first instruct LLM to generate questions for each group of procedures within the *attack sequence* to construct the *AttackSeq-Procedure-Yes*. Next, we randomly sample its 70% questions to generate the negative question samples. Specifically, we design two types of Yes-to-No transferring strategies as follows: (1) Negation of temporal prepositions, *i.e.*, changing "before" to "only after" and/or "after" to "only before", such that the modified question

Table 1: Evaluation results of four sub-tasks on human evaluation and automatic evaluation.

| Task | Human Evaluation | | | | | | | Automatic Evaluation | | | | | |
|------|-----------|------|------|------|------|------|----------|------|------|------|------|------|----------|
| | Hum. Perf. | Ans. | Cla. | Log | Rel. | Con. | Ans. Con. | Ans. | Cla. | Log | Rel. | Con. | Ans. Con. |
| Tactic | 0.51 | 4.30 | 4.36 | 4.45 | 4.56 | 4.46 | 4.44 | 4.52 | 4.65 | 4.79 | 4.84 | 4.65 | 4.76 |
| Technique | 0.71 | 4.09 | 4.21 | 4.40 | 4.45 | 4.44 | 4.41 | 4.10 | 4.40 | 4.62 | 4.63 | 4.39 | 4.59 |
| Procedure-Yes | 0.74 | 4.88 | 4.70 | 4.88 | 5.00 | 4.81 | 4.94 | 4.02 | 4.06 | 4.61 | 4.47 | 3.78 | 3.89 |
| Procedure-No | 0.56 | 4.55 | 4.84 | - | - | 4.82 | 4.66 | 3.29 | 3.66 | - | - | 2.77 | 3.25 |
| Average | 0.63 | 4.45 | 4.53 | 4.57 | 4.67 | 4.63 | 4.61 | 3.98 | 4.19 | 4.67 | 4.64 | 3.90 | 4.12 |

contradicts the given *attack sequence* (Rajpurkar et al., 2018); (2) Replacement of the procedures in the question with another procedures that is not supported by the given *attack sequence*.

**Q&A Refinement.** While LLMs possess remarkable text generation capabilities, these models may deviate from the requirements specified in users' instructions (Joshi et al., 2025), resulting in the conflict between the generated questions and the order of TTPs in *attack sequences*. Inspired by the Self-Refine framework (Madaan et al., 2023), we design a refinement criteria to iteratively refine the initial questions via the same LLM. To perform a holistic evaluation, we introduce six aspects below that emphasizes the question's linguistic (*i.e.,* Clarity) and task-oriented properties (Fu et al., 2024). Here, we divide the task-oriented aspects into three categories: (1) Question Complexity (*i.e.,* Answerability); (2) Content Alignment (*i.e.,* Relevance, Consistency, Answer Consistency); (3) Attack Sequence Alignment (*i.e.,* Logical) (*cf.* Appendix A.2). Considering the foundational role of Answerability in benchmark design, we first instruct the LLM to assess whether each question satisfies this criterion—specifically, whether the CTI report provides direct evidence supporting a correct answer that is clearly preferable to alternatives. Questions failing this requirement are discarded from the next step. Secondly, the LLM is instructed to evaluate the questions based on the remaining five aspects, providing a numerical score (out of five) and feedback for each aspect. Lastly, the LLM is prompted to refine the questions based on the feedback given (*cf.* Appendix C.2). We repeat this three-step process once more to improve the quality of the questions, the questions with full numerical scores are added to our final QA dataset.

After the Q&A refinement, the data volume of four sub-tasks in our **AttackSeqBench** reduce from 2,158/2,937/1,393/3,249 to 1,697/1,917/1,223/1,412, filtering out 35.82% of the original samples that cannot satisfy the defined selection criteria. Additionally, we further illustrate the top-10 ATT&CK tactics and techniques within our dataset in Figure 7(a) and 7(b) respectively. The most frequent tactic and technique in the figure is associated with a key objective of APTs, highlighting the relevance of our Q&A dataset in capturing *attack sequences* based on real-world cyber attacks.

## 2.3 DATASET EVALUATION

LLMs demonstrate strong potential in solving complex tasks, but they inevitably exhibit even severer hallucinations, which has become a widely recognized concern in the research community. To address this, we adopt a hybrid approach towards evaluating the quality of the constructed QA dataset using the criteria defined in our Q&A refinement (*cf.* Section 2.2). We design 5-point Likert scales for each of the evaluation criterion, where higher scores indicate better alignment.

**Human Evaluation.** We first randomly sample 35 questions from each sub-task to construct a question set for human evaluation. Three cybersecurity experts are then invited to answer and evaluate the quality of our Q&A dataset based on the six aspects defined in Section 2.2. Based on Table 1, we observe that the average Human Performance (abbreviated as *Hum. Perf.*) equals 0.63, suggesting that these questions is challenging and deducible even for individuals with domain expertise. Notably, *AttackSeq-Procedure-No* is derived from *AttackSeq-Procedure-Yes* through Yes-to-No transferring strategies, that is, its Logical and Relevance are inherently misaligned with the *attack sequence*, and we therefore do not evaluate these two aspects. Furthermore, the human evaluation shows consistently high average scores across all aspects, ranging from 4.45 to 4.61 out of 5, indicating that the generated Q&A are easy to comprehend and well aligned with the *attack sequences*.

**Automatic Evaluation.** To alleviate the laborious task of human evaluation, recent works (Zheng et al., 2023; Yao et al., 2024) have shown considerable effectiveness of LLM-as-a-Judge framework in aligning with human preferences within specific domain, including cybersecurity (Xu et al., 2024a). We incorporate G-Eval (Liu et al., 2023), a Chain-of-Thought (CoT) (Wei et al., 2022) and

form-filling paradigm, to systematically assess the quality of generated Q&A pairs. Specifically, we design individual prompts for each aspect in the evaluation criteria that includes its definition and the scoring guideline based on the same 5-point Likert scale in Human Evaluation (*cf.* Table 5 in Appendix A.2). Then we instruct GPT-4o rate the Q&A for each aspect based on the evaluation criterion and the correct answer from the ATT&CK KB (*cf.* Appendix C.3). Based on Table 1, we observe that *Logical* and *Relevance* are the highest rated aspects, reinforcing the LLM's ability to construct questions that follow the logical order of attack sequences. The fact that automatic evaluation scores are lower than human evaluation scores further indicates that answering questions correctly in our **AttackSeqBench** is more challenging for LLMs than for domain experts.

## 3 BENCHMARK AND EXPERIMENTS

### 3.1 BENCHMARK SETTINGS

As illustrated in Figure 3, we elaborate on three benchmark settings with varying levels of contextual information: (1) Zero-shot setting, (2) Context setting, and (3) RAG-empowered setting.

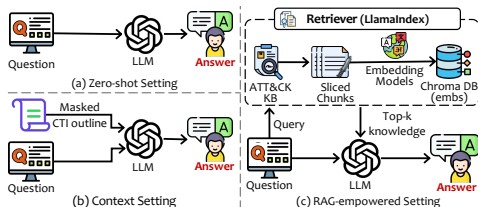

**Zero-shot Setting.** Motivated by the significant zero-shot reasoning abilities of LLMs across several downstream tasks (Hou et al., 2024; Kojima et al., 2022), we directly utilize the system prompt and the Q&A pairs to evaluate the LLMs' performance on three tasks based on their inherent knowledge.

Figure 3: Overview of the three benchmark settings that exhibit varying levels of contextual information given to the LLM.

**Context Setting.** Considering the existing context-aware work (Ma et al., 2023; Jin et al., 2024), we also organize this context setting to evaluate LLMs' Domain-Specific Epistemic Scalability at the semantic level. Here, we remove the corresponding summary of the ground truth tactic $t_k$ from the CTI outline $O$ to construct the *masked CTI outline* $O_m$, where $O_m = O \setminus o_k$. Afterwards, the LLM will be instructed to use the corresponding $O_m$ to answer the question, highlighting its potential to perform abductive reasoning to determine the most plausible TTP in *attack sequence*.

**RAG-empowered Setting.** Previous studies have demonstrated the Retrieval Augmented Generation (RAG) can significantly enhance the reliability pf LLMs and mitigate hallucinations (Zhang et al., 2025a). Here, we also design the RAG-empowered setting to evaluate the Domain-Specific Epistemic Scalability of LLMs at the representation level. This leverages the LLMs' in-context learning ability to learn the associations between the entity in the question's body and the relevant TTPs, thereby decomposing the problem and eliciting its stronger reasoning ability (Wu et al., 2022).

### 3.2 IMPLEMENTATION DETAILS

To investigate the CRU capability of existing models, we evaluate seven LLMs (*i.e.,* LLaMa-3.1-8B (Grattafiori et al., 2024), ChatGLM-4-9B (GLM et al., 2024), Qwen-2.5-3B, Qwen-2.5-14B, Qwen-2.5-32B (QwenTeam, 2024), Llama-3.3-70B (Grattafiori et al., 2024), and GPT-4o (OpenAI, 2024a) ) and five LRMs (*i.e.,* R1 (Llama-3.1-8B), R1(Qwen-2.5-14B), R1(Qwen-2.5-32B) (DeepSeek-AI, 2025), QWQ-32B (Team, 2024) and GPT-o3-mini (OpenAI, 2025)) on **AttackSeqBench**. We also utilize four post-training strategies (*i.e.,* SFT (Zhang et al., 2023), RD (Huang et al., 2024), RLIF (Zhao et al., 2025) and RLVR (DeepSeek-AI, 2025)) to embed security knowledge into LLMs to evaluate the Domain-Specific Epistemic Scalability of our **AttackSeqBench** (*cf.* Appendix A.4). Here, we measure the performance with accuracy $Acc = n/M$, where $n$ is the correctly-answered number of questions and $M$ is the total number. [1]

### 3.3 PERFORMANCE COMPARISON

**Comparison between diverse groups of LLMs.** As shown in Table 2, we notice that: Although LLMs generally follow the *scaling laws* in our **AttackSeqBench**, **none of the LLMs consistently outperforms the others**, and the optimal LLM varies diverse tasks. For instance, the best-

---

[1]Due to the page limitation, we introduce the complete implementation details in Appendix A.5 and A.6.

Table 2: Performance comparison of various LLMs, LRMs and post-training strategies across three benchmark tasks and settings. Bold and underlined denote best and second-best in each column.

| LLMs | AttackSeq-Tactic | | | AttackSeq-Technique | | | AttackSeq-Procedure | | |
|---|---|---|---|---|---|---|---|---|---|
| | Zero-Shot | Context | RAG | Zero-Shot | Context | RAG | Zero-Shot | Context | RAG |
| Qwen-2.5-3B | 0.4614 | 0.4467 | 0.3296 | 0.6121 | 0.5573 | 0.5249 | 0.5402 | 0.6037 | 0.4514 |
| Llama-3.1-8B | 0.5272 | 0.4897 | 0.4803 | 0.6355 | 0.6288 | 0.6077 | 0.5541 | 0.6845 | 0.5318 |
| ChatGLM-4-9B | 0.4806 | 0.4824 | 0.4588 | 0.6251 | 0.6359 | 0.6140 | 0.5481 | 0.6384 | 0.5327 |
| Qwen-2.5-14B | 0.5653 | 0.5928 | 0.5307 | 0.6891 | 0.6865 | **0.6987** | 0.6163 | 0.7063 | 0.6000 |
| Qwen-2.5-32B | **0.5903** | 0.6195 | 0.5154 | **0.7103** | **0.7267** | 0.6948 | 0.6269 | **0.7159** | 0.6024 |
| Llama-3.3-70B | 0.5643 | **0.6480** | 0.5394 | 0.6844 | 0.7022 | 0.6971 | 0.5483 | 0.6969 | 0.5342 |
| GPT-4o | 0.5710 | 0.5539 | 0.5522 | 0.6980 | 0.6041 | 0.6860 | 0.6767 | 0.5886 | 0.6319 |
| R1 (Llama-3.1-8B) | 0.4893 | 0.4474 | 0.4905 | 0.5526 | 0.5817 | 0.5740 | 0.5140 | 0.6278 | 0.5226 |
| R1 (Qwen-2.14B) | 0.5687 | 0.5219 | 0.5516 | 0.6105 | 0.6406 | 0.6286 | 0.6094 | 0.6911 | 0.5939 |
| R1 (Qwen-2.5-32B) | 0.5792 | 0.5938 | **0.5549** | 0.6265 | 0.6569 | 0.6395 | 0.6229 | 0.7055 | 0.6164 |
| QWQ-32B | 0.3439 | 0.5237 | 0.4712 | 0.3952 | 0.5224 | 0.5497 | 0.5746 | 0.7006 | 0.5566 |
| GPT-o3-mini | 0.5539 | 0.5274 | 0.5115 | 0.6051 | 0.5425 | 0.5853 | **0.6911** | 0.6850 | **0.6474** |
| Qwen-2.5-3B-Base | 0.2994 | 0.3424 | 0.4025 | 0.4997 | 0.5352 | 0.5848 | 0.0789 | 0.0862 | 0.4099 |
| SFT (Qwen-2.5-3B-Base) | 0.4479 | 0.4143 | 0.4063 | 0.5780 | 0.5550 | 0.5767 | 0.4706 | 0.5055 | 0.5321 |
| RD (Qwen-2.5-3B-Base) | 0.3866 | 0.3123 | 0.3536 | 0.5290 | 0.4564 | 0.4857 | 0.4945 | 0.4459 | 0.4812 |
| RLIF (Qwen-2.5-3B-Base) | 0.2434 | 0.1173 | 0.1962 | 0.5065 | 0.2869 | 0.3709 | 0.4873 | 0.4493 | 0.4619 |
| RLVR (Qwen-2.5-3B-Base) | 0.4396 | 0.3813 | 0.3689 | 0.5472 | 0.4987 | 0.5018 | 0.5237 | 0.5465 | 0.5199 |

performing models under the zero-shot setting across the three benchmark tasks are Qwen-2.5-32B, Qwen-2.5-32B, and GPT-o3-mini, respectively. This suggests that current models may not possess explicit security-specific knowledge, as relevant information in pretraining corpus is likely overshadowed by general-domain content. Moreover, most models consistently perform worst in *AttackSeq-Tactic* compared to the other two tasks, mirroring the human evaluation results in Section 2.3 and underscoring the common challenge faced by both human experts and LLMs in tactical inference.

Furthermore, from Table 3, we can observe that compared to the zero-shot setting, all models exhibit substantial performance gains on *AttackSeq-Procedure-No* under the context setting, indicating the importance of contextual information in identifying highly implausible actions within *attack sequences*. As defined in Appendix A.3, *AttackSeq-Procedure-No* is inherently more complex and reasoning-demanding than *AttackSeq-Procedure-Yes*, as it requires models to overcome the helpful-only bias and explicitly answer 'No' to disprove the plausibility of procedures occurring within the *attack sequence*. This explains why LRMs with stronger reasoning ability outperform in *AttackSeq-Procedure-No* compared to other tasks, underscoring the benchmark's emphasis on ***Reasoning Scalability***. Finally, most post-training strategies substantially improve the performance of its base LLM, particularly in zero-shot settings that rely solely on internal knowledge. However, their performance still lags behind instructive LLMs equipped with task-adapted prompts. This highlights a promising direction: Designing specialized post-training strategy to embed security-related knowledge into existing LLMs, thereby advancing the development of domain-specific models for cybersecurity.

**Comparison on Contextual Information.** Comparing the performance of LLMs across three benchmark settings in Table 2, we can observe that: In general, the Context setting consistently outperforms Zero-Shot and RAG settings across most benchmark tasks, with the advantage more pronounced in larger LLMs. Taking the Qwen-2.5 series as an example: performance shifts from zero-shot being optimal in Qwen-2.5-3B

Table 3: Performance comparison of two LLMs and two LRMs on *AttackSeq-Procedure-Yes* and *AttackSeq-Procedure-No* in each benchmark setting, where the bold value indicates the best performance of each column.

| LLMs | AttackSeq-Procedure-Yes | | | AttackSeq-Procedure-No | | |
|---|---|---|---|---|---|---|
| | Zero-Shot | Regular | RAG | Zero-Shot | Regular | RAG |
| Llama-3.1-8B | 0.9128 | 0.7572 | 0.8858 | 0.2434 | 0.6216 | 0.2111 |
| GPT-4o | **0.9469** | **0.9567** | 0.8831 | 0.4426 | 0.2698 | 0.4143 |
| R1 (Llama-8B) | 0.9332 | 0.8427 | **0.9191** | 0.1508 | 0.4417 | 0.1792 |
| GPT-o3-mini | 0.7612 | 0.7048 | 0.7408 | **0.6303** | **0.6678** | **0.5552** |

(0.4467 vs. **0.4614** vs. 0.3296) to context-setting being optimal in Qwen-2.5-32B (**0.6195** vs. 0.5903 vs. 0.5154) in the AttackSeq-Tactic task, with Qwen-2.5-14B showing the transition in between. This phenomenon is reasonable as larger LLMs possess more extensive internal knowledge, and task-specific context further enhances their effectiveness and robustness within the specific domain. Moreover, both LLMs and LRMs consistently fail to reach optimal performance under the RAG-empowered setting. This indicates that naive retrieval integration may introduce additional noise instead of enhancing results, underscoring the requirement for more advanced retrieval-augmented approaches. We further investigate its limitation in Section 3.5.2.

## 3.4 ROBUSTNESS ANALYSIS

### 3.4.1 PARAMETER SENSITIVITY ANALYSIS

Regarding the parameter sensitivity, we investigate the impact of temperature and maximum output tokens on LLMs and illustrate them in Figure 4(a) and Figure 4(b) respectively. Firstly, we observe that increasing the temperature from 0 to 1 causes a

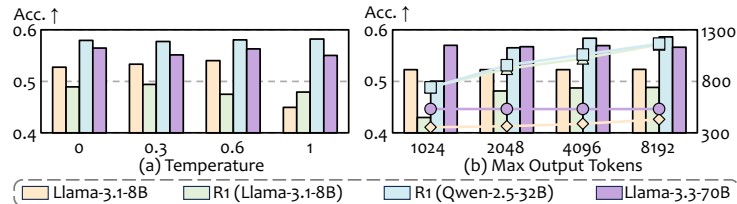

Figure 4: Parameter sensitivity analysis on (a) Temperature and (b) Max Output Tokens in *AttackSeq-Tactic* under the zero-shot setting.

sharp performance drop in smaller LLMs, while larger LLMs remain relatively unaffected in *attack sequence* analysis. This may be because smaller LLMs lack discriminative power and oscillate among suboptimal answers, whereas larger ones generate more stable logits that preserve correct outputs even under smoothing, while aligning with the observations of previous work in general domain (Renze, 2024; Li et al., 2025). On the other hand, we also releaze that increasing the token budget yields stable performance and output length for LLMs, whereas LRMs achieve significant gains in both performance and output tokens. Specifically, when the Max Output Tokens increase from 1,024 to 4,096, R1 (LLaMA-8B) and R1 (Qwen-32B) improve accuracy by 13.29% and 16.74%, with average output tokens increasing by 37.35% and 43.28%, respectively. However, when the token budget is further increased to 8,192, LRMs exhibit diminishing returns: average output tokens increase by 13.27% and 9.83%, while accuracy improves only by 0.29% and 0.46%. This highlights the importance of carefully tuning the maximum output tokens parameter to optimize performance in LRMs while considering the associated costs incurred (Wang et al., 2024).

### 3.4.2 COMPUTATIONAL COMPLEXITY ANALYSIS

As illustrated in Figure 5, we extensively compare the performance, model size and inference cost of several open-source LLMs and LRMs in *AttackSeq-Tactic* under the context setting. We can observe the following findings: (1) The performance trend of LLMs follows scaling law in our `AttackSeqBench` (Kaplan et al., 2020), particularly among LLMs within the same series; (2) Comparing the performance trend of LLMs with different thinking types, LRMs can not consistently outperform LLMs, yet they require significantly more inference time. This phenomenon is more pronounced when comparing LLMs and LRMs of the same model size, where distilled LRMs perform significantly worse than their base LLMs.

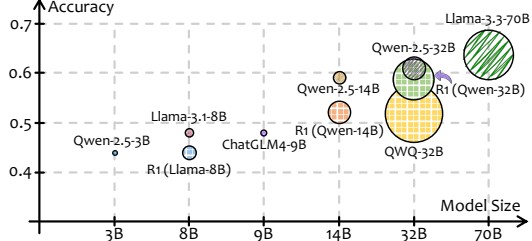

Figure 5: Computational complexity analysis of seven LLMs and four LRMs in *AttackSeq-Tactic* under the regular setting. The size of bubble represents inference time, where zigzag lines denote LLMs and cross hatch lines indicate LRMs.

This finding on inference time are consistent with existing studies in the general domain, whereas the performance results show a starkly contrasting trend (DeepSeek-AI, 2025). We further investigate this issue in Section 3.5.1.

## 3.5 IN-DEPTH ANALYSIS

### 3.5.1 REASONING PATH ANALYSIS

To explore the reason why existing LRMs perform worser than LLMs in *attack sequence* understanding, we provide a sample of reasoning path between Qwen-2.5-32B (LLM) and R1 (Qwen-2.5-32B) (LRM) on *AttackSeq-Procedure-No* under the Zero-shot setting in Table 4. We observe that both LLM and LRM can decompose the *attack sequence* into granular TTPs and retrieve relevant knowledge. Here, LLM successfully aligns the retrieved knowledge with the logical order of TTPs, thereby recognizing that generating network traffic before redirecting the entry point contradicts the tradi-

Table 4: A sample of reasoning path analysis between Qwen-2.5-32B (LLM) and R1(Qwen-2.5-32B) (LRM) on *AttackSeq-Procedure-No* under the zero-shot setting. We utilize different color to differentiate the diverse phases within a given *attack sequence*.

| **Question**: After retrieving the final payload from update.bokts[.]com, is it likely that CosmicStrand will interact with the Transport Device Interface to generate network traffic only **before** redirecting the entry point of the CSMCORE DXE driver to attacker-supplied code in the .reloc section during system startup? |
|---|
| **Answer Choices**: A) Yes  **B) No (Correct Answer)** |
| **Attack Sequence in Question**: Retrieve final payload → Interact with TDI → Redirect DXE |
| **Correct Attack Sequence**: Retrieve final payload → Redirect DXE→ Interact with TDI |

| LLM: Qwen-2.5-32B | LRM: R1 (Qwen-2.5-32B) |
|---|---|
| **Breakdown of Attack Sequence**: 1. The attacker retrieves the final payload from a specific domain (update.bokts[.]com). 2. The attacker interacts with the Transport Device Interface to generate network traffic. 3. The attacker redirects the entry point of the CSMCORE DXE driver to attacker-supplied code in the .reloc section during system startup. **Knowledge Retrieval**: 1. Retrieving the Final Payload: This is a common step in many malware attacks where the payload is fetched from a command and control (C2) server or a malicious domain. 2. Generating Network Traffic: Interacting with the Transport Device Interface to generate network traffic could be a technique used to exfiltrate data or to communicate with a C2 server. However, this step seems out of place if the primary goal is to redirect the entry point of a driver. 3. Redirecting the Entry Point: This is a sophisticated technique used in firmware or UEFI-based attacks to gain persistence and control over the system at a very low level. Generating network traffic (step 2) before redirecting the entry point (step 3) does not logically follow the typical sequence of such an attack. Final Answer: B: No **(Correct Conclusion)** | **Breakdown of Attack Sequence**: The question mentions retrieving a payload from a specific domain, update.bokts[.]com. Next, interacting with the Transport Device Interface (TDI) to generate network traffic. Then, during system startup, it redirects the entry point of the CSMCORE DXE driver to attacker-supplied code in the .reloc section. **Knowledge Retrieval**: TDI is part of the Windows kernel, used for handling network device drivers. If a malware interacts with TDI, it might be setting up a backdoor or a way to communicate over the network. DXE drivers are part of the UEFI (Unified Extensible Firmware Interface) system, which runs before the operating system starts. Usually, after gaining initial access (like downloading a payload), the next steps involve establishing persistence and communication. So, using TDI to set up network communication makes sense as a next step. **(Incorrect Initial Approach)** Wait **(Transitional Word)**, but the question says "only before redirecting." So, does the TDI interaction happen only before the redirection, meaning it's a one-time thing? Or is it part of the ongoing process? But wait **(Transitional Word)**, could the redirection happen before the network traffic? **(Questioning Alternatives)** I think I'm leaning towards yes, so the answer is A: Yes. **(Incorrect Conclusion)** |

tional attack sequence. It shows that LLM may rely on more direct sequence-matching between retrieved knowledge and procedural logic, enabling them to avoid unnecessary reasoning detours. In contrast, despite demonstrating reflective reasoning steps, LRM misinterprets the temporal constraint ("only before") and overemphasizes the plausibility of the TDI interaction. This overthinking within LRMs are also more prone to construct redundant reasoning loops and further incur reasoning misalignment, which may amplify minor misunderstandings into incorrect conclusions.

### 3.5.2 EFFECTIVENESS OF RAG STRATEGIES

To investigate why LLMs underperform in the RAG-empowered setting of our **AttackSeqBench**, we collect a candidate set where GPT-4o answers correctly in the zero-shot setting but fails under this setting. Specifically, we randomly sample 100 incorrect responses in *AttackSeq-Technique*, and classify them into four categories as shown in Figure 6. These four categories are: (1) *Factual Error*, meaning that LLM's prediction contradicts the ground truth despite the correct retrieved content; (2) *Over-reliance* (Xia et al., 2024), meaning that LLM excessively refers to the retrieved content and

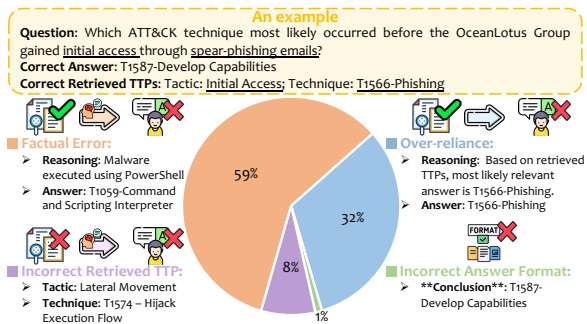

Figure 6: Error distribution of randomly-sampled 100 incorrect responses from GPT-4o in *AttackSeq-Technique* under the RAG-empowered setting.

fails to synthesize the *attack sequence* in the given question; (3) *Irrelevant Retrieved TTP*, which refers to incorrect predictions due to irrelevant retrieval to the given question; (4) *Incorrect Answer Format*, which refers to LLM's failure to follow the output format specified in the prompt template.

Our analysis reveals that 59% of errors stem from *Factual Error*, where the primary cause is the model's failure to effectively integrate retrieved evidence into the reasoning chain. Rather than enhancing the inference process, the retrieved knowledge functions as noise to distort the output distribution, thereby inducing faulty reasoning and incorrect answers. Moreover, around 32% of the errors occur because LLMs treat retrieved knowledge as the absolute authority without validating them against the question intent or their internal knowledge. Consequently, the model often relies solely on correct but incomplete retrieval chunks, which leads to faulty results. Within the ATT&CK KB, the nuances of TTP descriptions introduce several overlaps and ambiguities, which account for 8% of cases where the embedding model to retrieve incorrect tactics and techniques. For example,

technique *T1574 – Hijack Execution Flow* [2] is associated with three distinct tactics (*i.e., Persistence*, *Privilege Escalation*, and *Defense Evasion*), leading the model to misinterpret attack sequences in the given question. Enhancing the integration of retrieved knowledge with question intent and internal knowledge in RAG scenario, or investigating embedding methods capable of capturing fine-grained TTP semantics, holds promise for improving the effectiveness of *attack sequence* analysis.

## 4 RELATED WORK

**Automating CTI Report Understanding.** With the increasing demands of cybersecurity operations and the breakthrough of LLMs, researchers have progressively explored their applicability within CRU (Zhang et al., 2024a). For instance, prior works have showcased the remarkable capabilities of LLMs in interpreting TTPs from the ATT&CK KB, surpassing the performance of some of the fine-tuned LMs with cybersecurity data (Fayyazi et al., 2024). Meanwhile, another line of work proposes LLM-driven threat intelligence Knowledge Graph construction frameworks, which utilize the threat-related entities and relations to describe CTI reports in a structural manner (Huang & Xiao, 2024; Cheng et al., 2024). However, the extent to which LLMs can understand and reason about the precise relations between adversary behavioral sequences described in CTI reports remains largely under-explored. In our work, we perform a holistic evaluation on various pre-trained LLMs, LRMs and fine-tuned LLMs in *attack sequence* analysis, from deducing high-level tactics to detailed procedures described in CTI reports.

**Benchmarking LLMs in Cybersecurity.** Inspired by the remarkable open-world knowledge and complex inference ability within LLMs, various benchmarks have been proposed to evaluate its general capabilities in language understanding (Hendrycks et al., 2021b), math reasoning (Cobbe et al., 2021), code generation (Chen et al., 2021). Regarding the cybersecurity domain, researchers start to benchmark the abilities of LLMs under such specialized setting, such as ethical hacking and compliance (Liu, 2023; Tihanyi et al., 2024; Garza et al., 2023). Targeting CRU-related tasks, SEvenLLM (Ji et al., 2024) explores the abilities of LLMs in threat-related entities extraction and summarizing reports from security vendors. SecBench (Jing et al., 2025) evaluates the knowledge retention and logical reasoning abilities of existing pre-trained LLMs from multiple languages and dimensions. Meanwhile, CTIBench (Alam et al., 2024) introduces five benchmark tasks to explore the threat entity attribution and cause-tracing abilities of LLMs within the security context.

However, these studies primarily rely on authoritative sources (*e.g.,* textbooks, open standards) while overlooking real-world sources such as CTI reports. For instance, CTIBench solely incorporates a small-scale set of CTI reports in its dataset construction process for only one of its five benchmark tasks. Furthermore, these benchmarks remain insufficient for providing a comprehensive evaluation towards the LLMs' ability to understand relations among adversarial behaviors described in CTI reports, thereby failing to accurately reflect their reasoning capabilities over *attack sequences* containing domain-specific semantics. In this paper, we construct attack sequences based on an extensive set of CTI reports, while emphasizing on the practical aspects of CRU, inferring various aspects of adversarial behaviors, in our proposed benchmark tasks.

## 5 CONCLUSION

The breakthrough of LLMs has shown promising potential across the cybersecurity domain, particularly in CTI understanding. Despite this, the applicability of LLMs in analyzing adversarial sequences remains largely unexplored. In this work, we propose **AttackSeqBench**, a benchmark tailored for assessing LLMs' ability in understanding how adversaries operate through inferring TTPs based on *attack sequences* from real-world CTI reports. To cater to the evolving threat landscape, we design an automated Q&A construction pipeline that enables the Extensibility of our benchmark to new CTI reports. We further conduct extensive experiments across three settings with varying context availability, evaluating diverse LLMs, LRMs, and post-training strategies to verify its Reasoning Scalability and Domain-Specific Epistemic Expandability and thoroughly analyze their ability boundaries in *attack sequence* analysis. Our work opens up a new direction towards LLM-driven CRU, enabling effective threat intelligence mining through automation.

---

[2] https://attack.mitre.org/techniques/T1574/

ETHICS STATEMENT

Our work utilizes publicly available CTI reports, while ensuring that no proprietary information is used. The dataset generation pipeline is designed to maintain the integrity and accuracy of adversarial behavior sequences without fabricating or misrepresenting cyber threats. Furthermore, human evaluation is conducted with careful consideration of evaluator expertise and potential biases, ensuring fairness and reliability in assessment.

REPRODUCIBILITY STATEMENT

To promote reproducibility, we release an anonymous repository (https://anonymous.4open.science/r/AttackSeqBench) that contains all resources necessary to replicate our study, including the original CTI reports, the dataset construction pipeline, the complete datasets for the three tasks (*i.e., AttackSeq-Tactic*, *AttackSeq-Technique*, and *AttackSeq-Procedure*), and the code for running the three benchmark settings (*i.e.,* Zero-Shot setting, Context setting, and RAG-empowered setting). In addition, we also provide the comprehensive implementation details about the four post-training strategies in Appendix. Together, these resources ensure that our `AttackSeqBench` can be reliably reproduced and the reported results independently verified.

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

APPENDICES

APPENDIX TABLE OF CONTENTS

# A DATASET

## A.1 DATASET DISTRIBUTION

Based on Figure 7, we observe that the top three most frequent tactics (*i.e., Command and Control*, *Defense Evasion* and *Execution*) occur in the middle of attack sequences, while the bottom two tactics (*i.e., Exfiltration* and *Reconnaissance*) occurs at the start and the end of the attack sequence. Similarly, the most frequent ATT&CK technique is T1071-Application Layer Protocol [3], which is associated with the most common operations of APTs, *Command and Control*.

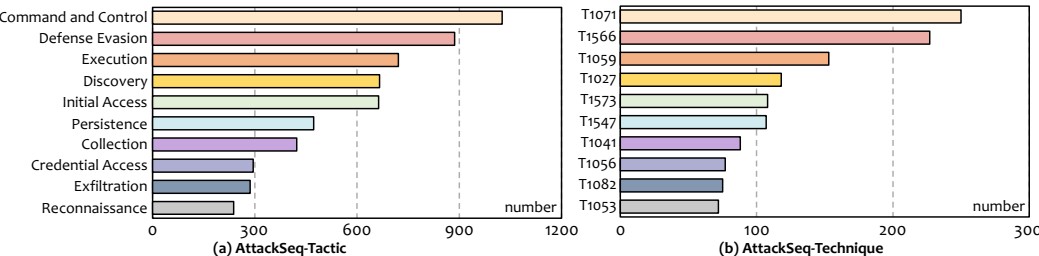

Figure 7: Visualization of the distribution of top-10 tactics and techniques in **AttackSeqBench**.

## A.2 DATASET EVALUATION CRITERIA

Inspired by the existing work (Fu et al., 2024), we utilize the following six dimensions as the evaluation criteria to evaluate the quality of our constructed Q&A dataset:

- *Answerability*. We check if there is direct evidence in the CTI outline that supports the correct answer, while clearly standing out as the best answer choice. Within this aspect, we also check if the correct answer can be inferred even if the associated summary to the correct answer's tactic is removed from the CTI outline.
- *Clarity*. We check if the question precise and unambiguous. More importantly, we also ensure that question avoid directly mentioning the correct answer such that the inference is required.
- *Logical*. We check if the sequence described in the question follow the order of tactics present in the attack sequence.
- *Relevance*. We check if the TTPs described in the question directly relate to the attack sequence.
- *Consistency*. We check if the question is consistent with the associated TTP that is used for question generation.
- *Answer Consistency*. We check if the question can be fully answered by the correct answer, without any contradictions and inconsistencies.

To quantitatively evaluate its quality, we first design the 5-point Likert scale for each aspect (refer to Table 5), where each score corresponds to a different level of the given aspect. Then we instruct three cybersecurity experts and LLM to provide the score of each aspect to achieve the human evaluation and the automatic evaluation, respectively. The detailed results are shown in Table 1. While the automatic evaluation results are lower than human evaluation, the human evaluation shows that most Q&A pairs in the dataset satisfy the requirements of all aspects. This suggests that automatic evaluation is still limited in knowledge-intensive domains such as in cybersecurity. Note that for the *AttackSeq-Procedure-No*, we evaluate questions only on four aspects—*Answerability*, *Clarity*, *Consistency*, and *Answer Consistency*—since it is derived from *AttackSeq-Procedure-Yes* through negation of temporal prepositions and replacement of procedures.

## A.3 BENCHMARK TASKS

Inspired by existing LLM benchmarks in the general domain (Hendrycks et al., 2021a; Rein et al., 2023; Zhang et al., 2024b), we propose three tasks in the form of Multiple-Choice Questions and Yes-No Questions to evaluate the reasoning capabilities of LLMs in inferring TTPs present in *attack sequences*, where each task reflects a distinct aspect of adversarial behaviors.

---

[3] https://attack.mitre.org/techniques/T1071/

Table 5: Annotation instructions for each evaluation aspect.

| Aspects | Instructions |
|---|---|
| Answerability | **Score 1:** The correct answer is not supported by the CTI outline. The information is either missing or contradicts the correct answer. Without the masked tactic paragraph, it is impossible to deduce the correct answer.
**Score 2:** Some evidence in the CTI outline loosely supports the correct answer, but it does not clearly stand out as the best choice. Removing the masked tactic paragraph makes it highly difficult to deduce the answer, even when referring to the MITRE ATT&CK KB.
**Score 3:** The correct answer has partial support in the CTI outline but is not explicitly stated. After removing the masked tactic paragraph, it is possible but challenging to infer the correct answer using the remaining information and MITRE ATT&CK KB.
**Score 4:** The correct answer is well-supported by the CTI outline and is the most reasonable choice based on the provided information. If the masked tactic paragraph is removed, the answer remains largely deducible using remaining information, and MITRE ATT&CK KB.
**Score 5:** The correct answer is directly supported by the CTI outline and is unambiguously the best choice. Even if the masked tactic paragraph is removed, the answer remains easily deducible based on the remaining CTI outline and MITRE ATT&CK KB. |
| Clarity | **Score 1:** The question is highly ambiguous, imprecise, or contains vague phrasing. It may directly state the correct answer, making inference unnecessary.
**Score 2:** The question is somewhat unclear or contains minor ambiguities. It may hint too strongly at the correct answer, reducing the need for inference.
**Score 3:** The question is fairly clear, but minor ambiguities exist. It does not directly state the correct answer, but slight rewording could improve precision.
**Score 4:** The question is mostly clear and unambiguous. It requires inference and does not directly reveal the correct answer.
**Score 5:** The question is precise, completely unambiguous, and free of vague phrasing. The correct answer is never directly mentioned, ensuring inference is required. |
| Logical | **Score 1:** The question does not align with the logical sequence of MITRE ATT&CK tactics in the CTI outline.
**Score 2:** The question shows minimal alignment with the MITRE ATT&CK sequence. It may reference unrelated tactics.
**Score 3:** The question has some logical alignment, but it may not reference preceding or subsequent tactics clearly.
**Score 4:** The question follows the sequence of MITRE ATT&CK tactics and references preceding or subsequent TTPs in a logical manner.
**Score 5:** The question perfectly aligns with the MITRE ATT&CK framework, referencing relevant TTPs in a way that naturally leads to the correct answer. |
| Relevance | **Score 1:** The question is completely unrelated to the CTI outline.
**Score 2:** The question has only slight relevance to the CTI outline but is mostly off-topic.
**Score 3:** The question is somewhat related to the CTI outline but could be refined to better fit the content.
**Score 4:** The question is directly related to the CTI outline, with minor room for improvement.
**Score 5:** The question fully aligns with the CTI outline and is highly relevant to the content. |
| Consistency | **Score 1:** The question contradicts the TTP description or is entirely misaligned with the provided details. |

Table 5 – continued from previous page

| Aspects | Instructions |
| --- | --- |
| **Consistency** | **Score 2:** The question loosely aligns with the TTP description but has inconsistencies or inaccuracies.
**Score 3:** The question mostly aligns with the TTP description but contains minor inconsistencies.
**Score 4:** The question is highly consistent with the TTP description, with only minor areas for improvement.
**Score 5:** The question fully aligns with the TTP description, with no inconsistencies or contradictions. |
| **Answer Consistency** | **Score 1:** The correct answer does not fully resolve the question, leaving contradictions or gaps.
**Score 2:** The correct answer provides some resolution, but contradictions or inconsistencies remain.
**Score 3:** The correct answer is mostly consistent, but minor contradictions exist.
**Score 4:** The correct answer fully resolves the question with minimal inconsistencies.
**Score 5:** The correct answer completely and unambiguously answers the question, with no contradictions or inconsistencies. |

**AttackSeq-Tactic**. This task evaluates the LLMs' capability to *infer a tactic* $t_k \in T$. Given a question $Q$ that corresponds to tactic $t_k$ and four shuffled candidate tactics $C_{Tac} = \{c_r : r \in [1,4]\}$, the LLM will be instructed to select the correct tactic $c_l \in C_{Tac}$.

**AttackSeq-Technique**. This task assesses the LLMs' capability to *infer a technique* $e_{j,k} \in E(t_k)$. Given a question $Q$ that corresponds to $e_{j,k}$ and four shuffled candidate techniques $C_{Tec} = \{c_r : r \in [1,4]\}$, the LLM will be instructed to select the correct technique $c_l \in C_{Tec}$.

**AttackSeq-Procedure**. This task challenges the LLMs' capability to *determine the likelihood of procedures* $p_{m,j,k} \in P(e_{j,k})$ in an *attack sequence*. Given a question $Q$ and two candidate choices $C_{Pro} = \{yes, no\}$, the LLM will be instructed to determine if the procedure $p_{m,j,k}$ is likely to occur in the given *attack sequence* $S$.

We further divide **AttackSeq-Procedure** into two sub-tasks, namely **AttackSeq-Procedure-Yes** and **AttackSeq-Procedure-No**, based on the ground truth of the boolean question. This explores the LLMs' ability in determining misleading procedures that are unlikely to occur in an *attack sequence*.

### A.4 BASELINES

To demonstrate the effectiveness and robustness of our proposed `AttackSeqBench`, we evaluate seven large language models, five large reasoning models and four post-training strategies across three tasks involving different levels of data and three benchmark settings with varying context completeness. We leverage vLLM (Kwon et al., 2023) to run all the open-source LLMs locally with two Nvidia H100 GPUs. For the colsed-source LLMs, we utilize OpenAI's Batch API [4] to conduct inference in batches. In our experiments, we set the following sampling parameters while keeping the default value for the remaining parameters: temperature to 0, maximum output tokens to 2048, and top_p to 1. Below is the details of the utilized LLMs, LRMs and post-training strategies in our experiments:

**Large Language Models:**

- **LLaMa-3.1-8B** (Grattafiori et al., 2024) is an instruction-tuned LLM from Meta, balancing performance and efficiency for textual understanding tasks.

- **ChatGLM-4-9B** (GLM et al., 2024) is pretrained on ten trillions of tokens and further achieve the high-quality alignment through supervised fine-tuning and human feedback learning.

---

[4] https://platform.openai.com/docs/guides/batch

- **Qwen2.5-3B**, **Qwen2.5-14B**, **Qwen2.5-32B** and **Qwen2.5-72B** (QwenTeam, 2024) represents Qwen-2.5 series LLMs with different parameter scales, demonstrating strong instruction-following and long-text generation capabilities.
- **LLaMa-3.3-70B** (Grattafiori et al., 2024) is an auto-regressive language model which is instruction-tuned in 70B with SFT and reinforcement learning with human feedback (RLHF).
- **GPT-4o** (OpenAI, 2024a) is one of the most advanced closed-source LLMs, which is a multi-lingual and multi-modal language model developed and functions well in real-time processing.

**Large Reasoning Models:**

- **DeepSeek-R1-Distill-Llama-8B (R1 (Llama-8B))**, **DeepSeek-R1-Distill-Qwen-14B (R1 (Qwen-14B))** and **DeepSeek-R1-Distill-Qwen-32B (R1 (Qwen-32B))** (DeepSeek-AI, 2025) are fine-tuned from the Llama-3.1-8B, Qwen2.5-14B and Qwen2.5-32B with 800k samples curated with DeepSeek-R1, aiming to equip smaller models with reasoning capabilities like DeepSeek-R1.
- **QWQ-32B-Preview (QWQ-32B)** (Team, 2024) is a preview release which gives LLM time to ponder, to question, and to reflect, enabling the deeper insight into complex problems.
- **GPT-o3-mini** (OpenAI, 2025) is designed with a focus on enhancing LLMs' reasoning capabilities. It leverages the Chain of Thought (CoT) to break down complex problems into several simpler steps to achieve this objective.

**Post-Training Strategies:**

- **Supervised Fine-tuning (SFT)** (Zhang et al., 2023) is a critical process for adapting pre-trained LLMs to specific tasks by training them on a task-specific dataset with labeled examples.
- **Reasoning Distillation (RD)** (Huang et al., 2024) RD is a widely adopted approach for enhancing LLM reasoning, which collects reasoning samples with self-reflection from existing LRMs and distills them to guide LLMs in acquiring long-thought capabilities.
- **Reinforcement Learning from Internal Feedback (RLIF)** (Zhao et al., 2025) replaces the external rewards in Group Relative Policy Optimization (GRPO) with LLMs' self-certainty, enabling unsupervised learning from intrinsic signals without relying on external rewards.
- **Reinforcement Learning with Verifiable Rewards (RLVR)** (DeepSeek-AI, 2025) leverages rule-based verification functions to provide reward signals for tasks with clear correctness criteria, enabling the optimization of LLMs while avoiding the complexities and potential pitfalls of reward models within RLHF.

### A.5 POST-TRAINING CORPUS CONSTRUCTION

Considering the post-training strategies, we construct two diverse datasets for SFT and RD, RLIF and RLVR respectively. For the former, we utilize a subset of the Primus-Instruct dataset (Yu et al., 2025). Primus-Instruct is a cybersecurity corpus collected for instruction-tuning, containing diverse task types such as alert explanation, suspicious command analysis, security event query generation, retrieved security document QA, Terraform security mis-configuration repair, and general multi-turn instruction following. To mitigate the inherent bias from linguistic inconsistencies, we filter out non-English samples via the FastText language identification library (Joulin et al., 2016) and manually verify the results, yielding a subset of 710 samples for SFT.

Regarding the latter, we use Primus-Reasoning, a cybersecurity reasoning distillation corpus constructed with DeepSeek-R1 (DeepSeek-AI, 2025) and GPT-o1-preview (OpenAI, 2024a). This dataset includes, but is not limited to, tasks such as Common Weakness Enumeration (CWE) mapping, Common Vulnerabilities and Exposures (CVE) analysis, and multiple-choice questions on general cybersecurity knowledge. Following (Zhang et al., 2025b), we leverage *transitional words* (*i.e.,* "but", "however", "wait", etc.) as the proxy for inferability, and retain only the 3,890 samples containing at least ten such words when constructing the corpus for RD, RLIF and RLVR.

### A.6 IMPLEMENTATION DETAILS

To examine the performance of LLMs on our **AttackSeqBench** after embedding cybersecurity knowledge, and considering GPU constraints, we evaluate existing post-training strategies on Qwen-2.5-3B (QwenTeam, 2024) and LLaMA-3.1-8B (Grattafiori et al., 2024) under both full-parameter fine-tuning and parameter-efficient fine-tuning paradigms across all benchmark tasks and settings.

**Retrieval Augmented Generation (RAG)**: We first crawl the description and the example procedures of each technique in the Enterprise ATT&CK Matrix v17 [5]. Then, we split the textual data into text chunks and embed the chunks in a vector store (*i.e.,* Chroma DB [6]), where each chunk's metadata contains the associated ATT&CK tactic and technique. We utilize a hybrid retriever with a re-ranker, by combining combine Okapi BM25 (Robertson & Zaragoza, 2009) and a dense retriever based on the more advanced text-embedding-3-large from OpenAI (OpenAI, 2024b). We set the chunk size to 512 and retrieved chunks to 3, and utilize LlamaIndex (Liu, 2022) to implement the retriever. Additionally, we also implement BGE-EN-ICL (Li et al., 2024) and ATT&CK-BERT (Abdeen et al., 2023a) within RAG to evaluate their effectiveness in *attack sequence* analysis.

**Supervised Fine-tuning (SFT)**: We fine-tune the backbone LLM on the first dataset within Appendix A.5 using the LLaMA-Factory (Zheng et al., 2024) framework. Specifically, we deliberately restricted SFT to one epoch with the learning rate of $3 \times 10^{-6}$, leveraging DeepSpeed ZeRO Stage-3 with CPU offload for memory efficiency. In our preliminary experiments, extending training process to multiple epochs led to noticeable degradation in the LLMs' general capabilities outside the cybersecurity domain. This effect can be attributed to "catastrophic forgetting", where continued exposure to a narrow corpus may overwrite its previously acquired broad knowledge. Thus, a single epoch struck a balance between adapting the LLM to the cybersecurity tasks while preserving its pre-trained general-purpose performance.

**Reasoning Distillation (RD)** refers to fine-tune LLM on a reasoning dataset distilled from the advanced LRMs (*i.e.,* DeepSeek-R1 () and GPT-o1-preview ()), which enables the smaller LLM to inherit the reasoning behaviors of the above LRMs. For RD, we fine-tune our backbone LLM on the latter dataset within Appendix A.5 with the same parameter settings in SFT.

**Reinforcement Learning with Verifiable Rewards (RLVR)** extends reinforcement learning by incorporating verifiable signals as rewards, such as correctness checks or logical consistency that can be programmatically validated. Specifically, we implement RLVR with Group Relative Policy Optimization (GRPO) (Shao et al., 2024) on the Volcano Engine Reinforcement Learning (verl) framework, where group-normalized rewards reduce variance and stabilize training. We conduct RLVR with the learning rate of $3 \times 10^{-6}$ using Fully Sharded Data Parallel (FSDP).

**Reinforcement Learning with Internal Feedback (RLIF)** (Zhao et al., 2025) enables LLMs to optimize policies using intrinsic signals without relying external supervision. In particular, it replaces the external rewards in GRPO with the self-certainty scores, which estimate the LLMs' confidence from its outputs, to enable fully unsupervised learning while maintaining the stability benefits of GRPO. Here, we conduct RLIF on the same latter dataset within Appendix A.5 using the verl framework, adopting the same training hyperparameters and FSDP set-up as in RLVR.

### A.7    RELATED BENCHMARKS COMPARISON

To demonstrate the uniqueness and novelty of our work, we illustrate the key differences between existing CTI-related benchmarks and our `AttackSeqBench` which significantly highlights attack sequence analyzing in Figure 1. Specifically, existing CTI-related benchmarks primarily focus on evaluating LLMs on three aspects: (1) *CTI Classification*, classifying malicious actions to known adversary behaviors (Alam et al., 2023); (2) *CTI Extraction*, extracting entities relevant to threat intelligence from the unstructured text (Bhusal et al., 2024); (3) *CTI Inference*, inferring the attributions of cyber attacks described in the real-world CTI reports (Alam et al., 2024). While these benchmarks preliminarily investigate the information extraction capabilities of LLMs within the CTI-related secnario, their ability to understand the *sequential patterns* of adversarial behavior remains largely unexplored. Besides, although KB (*i.e.,* MITRE ATT&CK® (Strom et al., 2018)) document real-world adversary behaviors through the pre-defined attack patterns, analyzing the patterns individually is insufficient to fully capture the progression of cyber attacks as listed in CTI reports. The sophisticated and stealthy nature of APTs requires a comprehensive understanding of how adversaries transition between the different attack phases, which are orchestrated as an attack sequence. This raises the need to consider the sequential characteristics of a cyber attack within the given CTI report.

---

[5] https://attack.mitre.org/versions/v17/matrices/enterprise/
[6] https://www.trychroma.com/

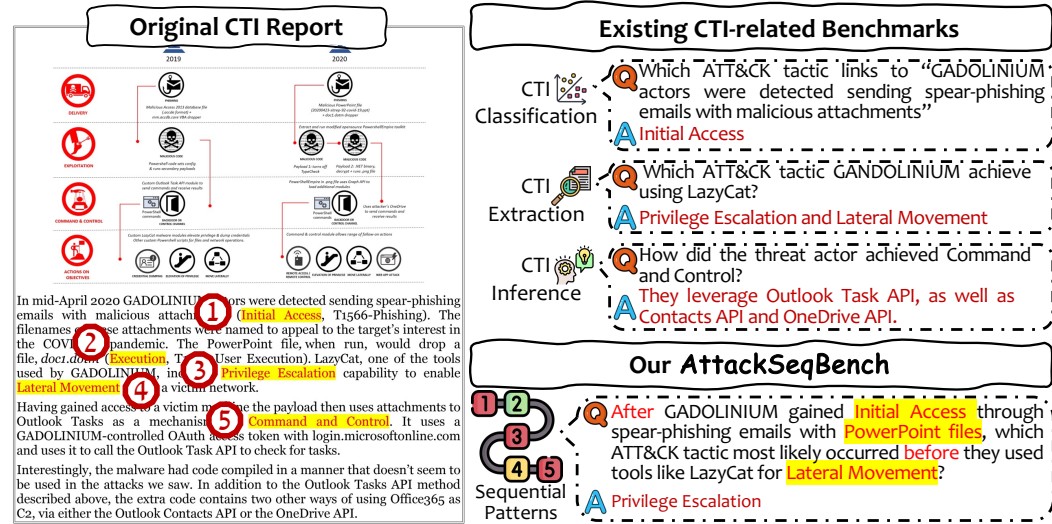

Figure 8: Comparison between existing CTI-related benchmarks and our **AttackSeqBench** based on a real-world CTI report (Koehl & Hannon, 2020). Our benchmark emphasizes on the sequences of adversary behaviors described in CTI reports.

## A.8 THE USE OF LARGE LANGUAGE MODELS

We declare that LLMs were employed to assist with the refinement of this manuscript, specifically for grammar checking and language polishing. Additionally, LLMs were used in a limited capacity for minor debugging and syntactic correction of code snippets. Beyond these auxiliary roles, given that the primary purpose of this work is to explore the capability of diverse types of LLMs in understanding adversarial *attack sequences*, we also utilize a range of open-source and closed-source LLMs and LRMs during dataset construction, dataset refinement, and performance evaluation. All such uses are documented in the main paper and the appendix and were carefully controlled to ensure transparency and reproducibility.

## A.9 LIMITATIONS AND FUTURE WORKS

**Limitations:** While our work serves as a pioneering study into the LLMs' reasoning capabilities in *attack sequence* analysis, several limitations should be acknowledged. Firstly, our study focuses on correctness of models' responses through Multi-Choice Questions and Yes-No Questions, which may not fully capture the reasoning abilities of LLMs necessary for comprehensive evaluation. Secondly, although we have conducted extensive experiments with seven LLMs, five LRMs, and four post-training strategies across three benchmark tasks (*AttackSeq-Tactic*, *AttackSeq-Technique*, and *AttackSeq-Procedure*) and three benchmark settings (Zero-Shot setting, Context setting, and RAG-empowered setting), fully demonstrating the Reasoning Scalability and Domain-Specific Epistemic Expandability of our **AttackSeqBench**, the implementations of RAG and post-training strategies remains relatively basic and leave room for future refinement. Thirdly, our **AttackSeqBench** currently leverages 408 rigorously filtered CTI reports to extract *attack sequences* and generate Q&A pairs. Although this number substantially exceeds prior CTI-related studies (*i.e.,* 12 in AttacKG+ (Zhang et al., 2025c), 12 in MM-AttacKG (Zhang et al., 2025d), and at most 71 in Attack Flow [7]), the proposed dataset construction pipeline is flexible and can be readily extended to unseen CTI reports. This not only demonstrates the Extensibility of our **AttackSeqBench** but also highlights an important direction for continuously refining this benchmark in future work. Nevertheless, while it is important to be aware of these limitations, our **AttackSeqBench** serves are a valuable benchmark to systematically explore LLMs' reasoning abilities across the tactical, technical and, procedural dimensions of adversarial behaviors.

---

[7]https://center-for-threat-informed-defense.github.io/attack-flow/

**Future works:** Building on the limitations, our future research on **AttackSeqBench** will proceed along three directions. Regarding evaluation, we plan to expand our evaluation methods from the simple Multiple-Choice Question tasks and Yes-No Question tasks to the more complex reasoning and completion tasks, thereby providing a more comprehensive assessment of model capabilities in CTI report understanding. In terms of methodology, we will build on **AttackSeqBench** to explore more fine-grained RAG approaches and advanced post-training strategies that account for the knowledge-extensive and high-stakes nature of CTI reports understanding, aiming to fully leverage model potential in complex cyber-attack scenarios. At the data level, we will continue to expand and dynamically update the CTI corpus to ensure our **AttackSeqBench** remains evolvable over time, thereby supporting the steady advancement of domain-specific foundation models for cybersecurity.

## B EXPERIMENTS

### B.1 TTP TEMPORAL POSITION ANALYSIS

To better understand the LLMs' capabilities in *attack sequence* analysis, we conduct fine-grained analysis based on each stage within the *attack sequences* of MITRE ATT&CK®. We illustrate the performance of two LLMs and two LRMs across all benchmark tasks in the Regular setting in Figure 9 and show the corresponding values, the mean and standard deviation (SD) of these LLMs and LRMs on each tactic and benchmark task in Table 6. We identify four overachieving attack phases to categorize the ATT&CK tactics in attack sequences: (1) Initial Intrusion Phase; (2) Exploitation Phase; (3) Stealth Expansion Phase; (4) Objective Orchestration Phase. It is worth noting that our categorization follows Tactics, as each Technique and Procedure in MITRE ATT&CK® is uniquely mapped to a specific Tactic within a given *attack sequence*.

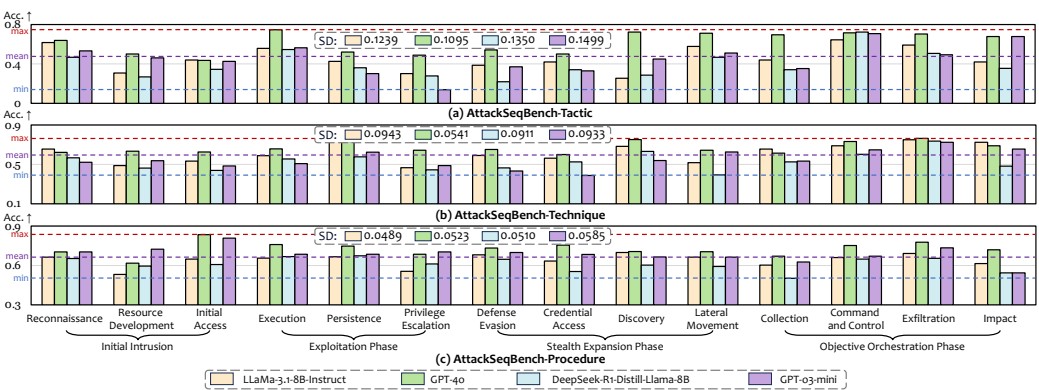

Figure 9: The performance comparison between two LLMs and two LRMs on each tactic in each benchmark task.

Table 6: Results of performance of two LLMs and two LRMs on all the 14 tactics, and the corresponding statistics of mean and standard deviation.

| | Tactic | Reconna-issance | Resource Devel. | Initial Access | Execu. | Persist. | Privilege Escala. | Defense Evasion | Credent. Access | Discov. | Lateral Move. | Collec. | Cmd. & Control | Exfiltr. | Impact | Mean | SD |
|---|---|---|---|---|---|---|---|---|---|---|---|---|---|---|---|---|---|
| Tactic | Llama-3.1-8B | 0.6170 | 0.3077 | 0.4406 | 0.5573 | 0.4260 | 0.3023 | 0.3867 | 0.4211 | 0.2551 | 0.5778 | 0.4390 | 0.6444 | 0.5915 | 0.4194 | 0.4561 | 0.1239 |
| | GPT-4o | 0.6383 | 0.5000 | 0.4368 | 0.7470 | 0.5207 | 0.4884 | 0.5430 | 0.5000 | 0.7245 | 0.7111 | 0.6951 | 0.7155 | 0.7042 | 0.6774 | 0.6144 | 0.1095 |
| | R1 (Llama-8B) | 0.4681 | 0.2692 | 0.3448 | 0.5455 | 0.3609 | 0.2791 | 0.2188 | 0.3421 | 0.2857 | 0.4667 | 0.3415 | 0.7238 | 0.5070 | 0.3548 | 0.3934 | 0.1350 |
| | GPT-o3-mini | 0.5319 | 0.4615 | 0.4269 | 0.5635 | 0.3018 | 0.1395 | 0.3711 | 0.3289 | 0.4490 | 0.5111 | 0.3537 | 0.7071 | 0.493 | 0.6774 | 0.4512 | 0.1499 |
| | Mean | 0.5638 | 0.3846 | 0.4123 | 0.6033 | 0.4024 | 0.3023 | 0.3799 | 0.3980 | 0.4286 | 0.5667 | 0.4573 | 0.6977 | 0.5739 | 0.5323 | - | - |
| | SD | 0.0786 | 0.1132 | 0.0454 | 0.0961 | 0.0938 | 0.1434 | 0.1325 | 0.0792 | 0.2149 | 0.1066 | 0.1643 | 0.0362 | 0.0971 | 0.1697 | - | - |
| Technique | Llama-3.1-8B | 0.6556 | 0.4878 | 0.5327 | 0.5890 | 0.7324 | 0.4667 | 0.5914 | 0.5625 | 0.6839 | 0.5179 | 0.6557 | 0.6884 | 0.7500 | 0.7241 | 0.6170 | 0.0943 |
| | GPT-4o | 0.6222 | 0.6341 | 0.6262 | 0.6568 | 0.7254 | 0.6444 | 0.6512 | 0.6000 | 0.7513 | 0.6429 | 0.6148 | 0.7329 | 0.7632 | 0.6897 | 0.6682 | 0.0541 |
| | R1 (Llama-8B) | 0.5667 | 0.4634 | 0.4393 | 0.5551 | 0.5775 | 0.4444 | 0.4651 | 0.5250 | 0.6321 | 0.3929 | 0.5246 | 0.6027 | 0.7368 | 0.4828 | 0.5292 | 0.0911 |
| | GPT-o3-mini | 0.5222 | 0.5366 | 0.4836 | 0.5085 | 0.6241 | 0.4889 | 0.4333 | 0.3875 | 0.5389 | 0.6250 | 0.5328 | 0.6473 | 0.7237 | 0.6552 | 0.5505 | 0.0933 |
| | Mean | 0.5917 | 0.5305 | 0.5205 | 0.5774 | 0.6649 | 0.5111 | 0.5353 | 0.5188 | 0.6516 | 0.5447 | 0.5820 | 0.6678 | 0.7434 | 0.6380 | - | - |
| | SD | 0.0591 | 0.0755 | 0.0802 | 0.0624 | 0.0764 | 0.0907 | 0.1031 | 0.0927 | 0.0896 | 0.1153 | 0.0638 | 0.0557 | 0.0170 | 0.1072 | - | - |
| Procedure | Llama-3.1-8B | 0.6634 | 0.5319 | 0.6489 | 0.6552 | 0.6667 | 0.5556 | 0.6809 | 0.6331 | 0.6987 | 0.6633 | 0.6027 | 0.6606 | 0.6906 | 0.6140 | 0.6404 | 0.0489 |
| | GPT-4o | 0.7030 | 0.6170 | 0.8351 | 0.7586 | 0.7469 | 0.6852 | 0.7325 | 0.7554 | 0.7067 | 0.7041 | 0.6712 | 0.7515 | 0.777 | 0.7193 | 0.7260 | 0.0523 |
| | R1 (Llama-8B) | 0.6535 | 0.5957 | 0.6064 | 0.6681 | 0.6728 | 0.6111 | 0.6474 | 0.5540 | 0.6027 | 0.5918 | 0.5023 | 0.6485 | 0.6547 | 0.5439 | 0.6109 | 0.0510 |
| | GPT-o3-mini | 0.7030 | 0.7234 | 0.8085 | 0.6853 | 0.6852 | 0.7037 | 0.6991 | 0.6835 | 0.6640 | 0.6633 | 0.6256 | 0.6707 | 0.7338 | 0.5439 | 0.6852 | 0.0585 |
| | Mean | 0.6807 | 0.6170 | 0.7247 | 0.6918 | 0.6929 | 0.6389 | 0.6900 | 0.6565 | 0.6680 | 0.6556 | 0.6005 | 0.6828 | 0.7140 | 0.6053 | - | - |
| | SD | 0.0260 | 0.0796 | 0.1139 | 0.0462 | 0.0368 | 0.0684 | 0.0355 | 0.0848 | 0.0473 | 0.0467 | 0.0714 | 0.0467 | 0.0530 | 0.0829 | - | - |

We summarize our findings into three key points: (1) Comparing the performance of three benchmark tasks, the overall highest standard deviation occurs within the *AttackSeq-Tactic* task for all LLMs and LRMs, suggesting that the LLMs' and LRMs' performance has the largest variance in this taskand further highlighting the challenge of inferring tactics within *attack sequences*; (2) In contrast, the standard deviation for *AttackSeq-Procedure* is the lowest for all LLMs, suggesting more consistency in the LLMs' and LRMs' performance among the different *attack sequence* stages; (3) GPT-4o consistently outperforms other LLMs and LRMs across most benchmark tasks, particularly in the Exploitation Phase and the Stealth Expansion Phase. However, the remaining LLMs achieve comparable performance in the remaining two phases, especially in tactics at the start and end of the

Table 7: Error analysis of GPT-4o in *AttackSeq-Technique*, where GPT-4o incorrectly associates technique T1574-Hijack Execution Flow with the Execution tactic.

| Q & A | Cases | LLM Response |
|-------|-------|--------------|
| **Question:** Before establishing persistence through creating a Run key in the registry, which ATT&CK technique most likely facilitated the execution of the backdoor FinalBot.exe after the WMI discovery process identified the presence of antivirus software? | Wrong | **A: T1574-Hijack Execution Flow** This technique involves hijacking the execution flow of a legitimate process to execute malicious code. DLL sideloading is a common method for this technique. The CTI report explicitly mentions that Sidewinder used DLL sideloading ('cmpbk32.dll' to execute 'Srvstr.exe'), which aligns with this technique. This likely facilitated the execution of the backdoor 'FinalBot.exe' after the WMI discovery process.
**B: T1003.001-LSASS Memory**: This technique involves dumping credentials from the LSASS process memory. There is no mention in the CTI report of credential dumping or any activity related to LSASS. This choice is irrelevant.
**C: T1059.008-Network Device CLI**: This technique involves using a command-line interface (CLI) to interact with network devices. The CTI report does not mention any interaction with network devices or the use of a CLI. This choice is irrelevant.
**D: T1129-Shared Modules**: This technique involves executing malicious code by loading shared modules. While DLL sideloading involves shared modules, the specific ATT&CK technique for DLL sideloading is T1574-Hijack Execution Flow. This choice is less precise than option A.
Thus, the most plausible sequence of TTPs involves the use of **T1574-Hijack Execution Flow** (via DLL sideloading) to execute the backdoor 'FinalBot.exe' after the WMI discovery process and before establishing persistence. **(Wrong Conclusion)** |
| **Options:** A) T1574-Hijack Execution Flow B) T1003.001-LSASS Memory C) T1059.008-Network Device CLI D) T1129-Shared Modules **(Correct Answer)** | Right | **A: T1574-Hijack Execution Flow**: This technique involves exploiting or redirecting the flow of execution to execute untrusted code. While this could potentially execute malware, it doesn't directly relate to the sequence of WMI discovery leading to the execution of 'FinalBot.exe'.
**B: T1003.001-LSASS Memory**: This technique involves accessing credentials in memory by dumping the LSASS process. This is unrelated to executing a backdoor like 'FinalBot.exe' after WMI discovery.
**C: T1059.008-Network Device CLI**: This technique involves command-line interfaces for network devices, which is unrelated to executing programs on a system post-discovery of antivirus software.
**D: T1129-Shared Modules**: This technique involves executing malicious payloads using shared libraries or modules. If 'FinalBot.exe' were executed via shared modules, this process could occur after determining system defenses, making it relevant to the scenario.
Answer: **D: T1129-Shared Modules**. This technique likely facilitated the execution of 'FinalBot.exe' after WMI discovery detected antivirus presence, allowing for a subtle execution method bypassing basic defenses. **(Correct Conclusion)** |

*attack sequences* (*e.g.,* Reconnaissance and Impact). This suggests that TTPs within rarely observed tactics are more challenging for the LLM and LRMs to interpret.

## B.2 CASE STUDY

To better understand the superiority and limitation of the advanced LLMs in analyzing *attack sequences*, we run GPT-4o multiple times to collect both successful and failed responses to the same questions under the identical experimental setting, which are presented in Table 7. For the *Wrong* case, GPT-4o incorrectly associates the technique of choice A, *i.e.,* T1574-Hijack Execution Flow, with the Execution tactic. Although this technique indeed involves the execution of malicious payloads based on the ATT&CK knowledge base, it is only associated to the following three tactics: Persistence, Privilege Escalation, and Defense Evasion. This incorrect interpretation suggests that GPT-4o struggles in distinguishing the inherent ambiguity found in TTP descriptions, thereby affecting their ability to analyze *attack sequences*. Regarding the *Right* one, GPT-4o correctly identifies T1129-Shared Modules as the most plausible technique, which belongs to the *Execution* tactic [8] and serves as the executable files that are loaded into processes to provide access to execute malicious payloads. By selecting this option, GPT-4o demonstrates its ability to reason over the *attack sequence*: after WMI discovery detects the presence of antivirus, shared modules would facilitate the execution of "FinalBot.exe" to bypass basic defenses. This correct interpretation is beneficial to effectively link the ambiguous textual cues with the appropriate tactic/technique entities, thereby improving its reliability in analyzing *attack sequences*.

## B.3 IMPACT OF EMBEDDING MODELS WITHIN RAG-EMPOWERED SETTING

The semantics within CTI reports contain a large volume of domain-specific technical terminologies, where the accuracy of retrievers in identifying the most relevant tactics, techniques and procedures critically influences LLMs' performance in RAG-empowered settings. To further examine this issue and mitigate the potential knowledge bias introduced by BGE-EN-ICL, we incorporate two

---

[8]https://attack.mitre.org/techniques/T1129/

Table 8: Performance comparison between three embedding models (abbreviate as Emb. M.).

| #Emb. M. Tasks | BGE | | | OPENAI | | | ATT&CK-BERT | | |
|---|---|---|---|---|---|---|---|---|---|
| | Tactic | Technique | Procedure | Tactic | Technique | Procedure | Tactic | Technique | Procedure |
| Llama-3.1-8B | 0.4751 | 0.5974 | 0.5243 | 0.4838 | 0.6103 | 0.5435 | 0.4820 | 0.5980 | 0.5334 |
| GPT-4o | **0.5522** | **0.6860** | 0.6319 | **0.5616** | **0.6578** | **0.6482** | **0.5687** | **0.7016** | 0.6216 |
| R1 (Llama-8B) | 0.4905 | 0.5740 | 0.5226 | 0.4932 | 0.5696 | 0.5150 | 0.4651 | 0.5804 | 0.5104 |
| GPT o3-mini | 0.5115 | 0.5853 | **0.6474** | 0.5192 | 0.5827 | 0.6474 | 0.5245 | 0.5874 | **0.6414** |

additional embedding models into our RAG-empowered setting, namely OpenAI's text-embedding-3-large (OpenAI, 2024b) and a domain-adapted ATT&CK BERT (Abdeen et al., 2023b) fine-tuned on the specific cybersecurity data. We conduct performance comparisons between two representative LLMs and two LRMs based on these above embedding models in Table 8. We observe that the three embedding models exhibit comparable performance across different benchmark tasks and settings, with only marginal differences. Among them, BGE-EN-ICL proves to be the most cost-efficient, generalizable, and effective choice, and thus we primarily report model performance based on this embedding throughout the paper. Notably, although ATT&CK-BERT contains far fewer parameters than BGE-EN-ICL (110M vs. 7B), it achieves comparable performance, underscoring the importance of injecting domain-specific security knowledge into LLMs and pointing to a promising direction for future work.

## C PROMPT TEMPLATES

### C.1 QUESTION GENERATION PROMPT TEMPLATES

We utilize few-shot prompt templates for question generation of each of our benchmark tasks (*i.e., AttackSeq-Tactic*, *AttackSeq-Technique*, *AttackSeq-Procedure-Yes*, *AttackSeq-Procedure-No*), the corresponding prompts are shown in Box 1, Box 2, Box 3, Box 4 respectively.

---

**Box 1: Few-shot prompt for question generation for *AttackSeq-Tactic*.**

You are a cybersecurity expert with deep knowledge of Cyber Threat Intelligence (CTI) reports and the MITRE ATT&CK framework.
**[Inputs]:**
You will receive two parts:
1. A CTI Report that describe a cyber attack ordered by MITRE ATT&CK tactics. Note that additional information labeled as "Others" provides context about the threat actor but is secondary.
2. A MITRE ATT&CK Tactic present in the CTI report.
**[End of Inputs]**

Your task is to generate a question about the attack sequence based on the MITRE ATT&CK tactics found in the CTI report, where the answer to the question is the given MITRE ATT&CK tactic. The question should focus on inferring the given tactic by using the attack sequence based on the remaining tactics in the CTI report.
**[Steps]:**
Please follow these steps:
1. Analyze the CTI report:
- Read the report carefully.
- Identify and list the attack sequence in the order presented by the MITRE ATT&CK tactics.
2. Construct the Question:
- Design a question that emphasizes the order of the attack sequence in the CTI report.
- The question should exclude the TTPs under the given MITRE ATT&CK tactic that are described in the CTI report. Instead, include the TTPs in the tactic that precedes before and/or follows after the given MITRE ATT&CK tactic based on the order of tactics in the CTI report.
- Ensure that the answer to the question is the given MITRE ATT&CK tactic.
- The question should be concise, clear, and targeted towards experienced cybersecurity professionals.
- Please refer to the example questions below for guidance.
**[Examples]:**
Example Questions:
- Question: After gaining initial access through compromised VPN accounts, which ATT&CK tactic most likely occurred before Ke3chang achieved persistence by adding a Run key? Answer: Execution
- Question: Which ATT&CK tactic most likely occurred before Axiom gained initial access to the victim's network using SQL injection? Answer: Resource Development
- Question: Which ATT&CK tactic most likely occurred after Ke3chang establishes connection with the C2 server through Internet Explorer (IE) by using the COM interface IWebBrowser2? Answer: Exfiltration
- Question: After using stolen code signing certificates to sign DUSTTRAP malware and components, which ATT&CK tactic most likely occurred before APT41 used Windows Services with names such as Windows Defend for persistence of DUSTPAN? Answer: Execution
**[End of Examples]**
3. Provide the Question-Answer Pair:
- Please follow the output format:
"Question: <insert question here> Answer: <insert answer here>"
**[End of Steps]**
Following the steps above, please generate a question based on the CTI report and ATT&CK tactic given below.

---

1404
1405
1406
1407
1408
1409
1410
1411
1412
1413
1414
1415
1416
1417
1418
1419
1420
1421
1422
1423
1424
1425
1426
1427
1428
1429
1430
1431
1432
1433
1434
1435
1436
1437
1438
1439
1440
1441
1442
1443
1444
1445
1446
1447
1448
1449
1450
1451
1452
1453
1454
1455
1456
1457

---

**Box 2: Few-shot prompt for question generation for *AttackSeq-Technique*.**

You are a cybersecurity expert with deep knowledge of Cyber Threat Intelligence (CTI) reports and the MITRE ATT&CK framework.

**[Inputs]:**

You will receive three parts:

1. A CTI Report that describe a cyber attack ordered by MITRE ATT&CK tactics. Note that additional information labeled as "Others" provides context about the threat actor but is secondary.

2. A MITRE ATT&CK Tactic present in the CTI report.

3. A MITRE ATT&CK Technique present in the CTI report.

**[End of Inputs]**

Your task is to generate a question about the attack sequence based on the MITRE ATT&CK tactics found in the CTI report, where the answer to the question is the given MITRE ATT&CK technique that belongs to the given ATT&CK tactic. The question should focus on inferring the given technique by using the attack sequence based on the remaining tactics in the CTI report.

**[Steps]:**

Please follow these steps:

1. Analyze the CTI report:

- Read the report carefully.

- Identify and list the attack sequence in the order presented by the MITRE ATT&CK tactics.

2. Construct the Question:

- Design a question that emphasizes the order of the attack sequence in the CTI report.

- The question should exclude the TTPs under the given MITRE ATT&CK tactic that are described in the CTI report. Instead, include the TTPs in the tactic that precedes before and/or follows after the given MITRE ATT&CK tactic based on the order of tactics in the CTI report.

- Ensure that the answer to the question is the given MITRE ATT&CK technique.

- The question should be concise, clear, and targeted towards experienced cybersecurity professionals.

- Please refer to the example questions below for guidance.

**[Examples]:** Example Questions:

- Question: After gaining initial access through compromised VPN accounts, which ATT&CK technique most likely occurred before Ke3chang achieved persistence by adding a Run key? Answer: T1059-Command and Scripting Interpreter

- Question: Which ATT&CK technique most likely occurred before Axiom gained initial access to the victim's network using SQL injection? Answer: T1583.002-DNS Server

- Question: Which ATT&CK technique most likely occurred after Ke3chang establishes connection with the C2 server through Internet Explorer (IE) by using the COM interface IWebBrowser2? Answer: T1020-Automated Exfiltration

- Question: After using stolen code signing certificates to sign DUSTTRAP malware and components, which ATT&CK technique most likely occurred before APT41 used Windows Services with names such as Windows Defend for persistence of DUSTPAN? Answer: T1569.002-Service Execution

**[End of Examples]**

3. Provide the Question-Answer Pair:

- Please follow the output format:

"Question: <insert question here> Answer: <insert answer here>".

**[End of Steps]**

Following the steps above, please generate a question based on the CTI report and ATT&CK tactic and technique given below.

**Box 3: Few-shot prompt for question generation for *AttackSeq-Procedure-Yes*.**

You are a cybersecurity expert with deep knowledge of Cyber Threat Intelligence (CTI) reports and the MITRE ATT&CK framework.

**[Inputs]:**

You will receive four parts:

1. A CTI Report that describe a cyber attack ordered by MITRE ATT&CK tactics. Note that additional information labeled as "Others" provides context about the threat actor but is secondary.

2. A MITRE ATT&CK Tactic present in the CTI report.

3. A MITRE ATT&CK Technique present in the CTI report.

4. 4. A list of Procedures present in the CTI report, where each procedure is represented as a (Subject, Relation, Object) triplet.

**[End of Inputs]**

Your task is to generate a question about the attack sequence based on the MITRE ATT&CK tactics found in the CTI report, the question should focus on inferring the given list of procedures based on the given MITRE ATT&CK tactic and technique. The answer to the question is "Yes", indicating that the given list of procedures is likely to occur in the attack sequence.

**[Steps]:**

Please follow these steps:

1. Analyze the CTI report:

- Read the report carefully.

- Identify and list the attack sequence in the order presented by the MITRE ATT&CK tactics.

2. Construct the Question:

- Design a question that emphasizes the order of the attack sequence in the CTI report.

- The question should exclude the TTPs under the given MITRE ATT&CK tactic that are described in the CTI report. Instead, include the TTPs in the tactic that precedes before and/or follows after the given MITRE ATT&CK tactic based on the order of tactics in the CTI report.

- Ensure that the answer to the question is "Yes".

- The question should be concise, clear, and targeted towards experienced cybersecurity professionals.

- Please refer to the example questions below for guidance.

**[Examples]:** Example Questions:

- Question: After gaining initial access through compromised VPN accounts, is it likely that the Ke3chang malware will run commands on the command-line interface before achieving persistence by adding a Run key? Answer: Yes

- Question: Is it likely that Axiom will acquire dynamic DNS services for use in the targeting of intended victims before gaining initial access to the victim's network using SQL injection? Answer: Yes

- Question: Is Ke3chang likely to perform frequent and scheduled data exfiltration from compromised networks after establishing connection with the C2 server through Internet Explorer (IE) by using the COM interface IWebBrowser2? Answer: Yes

- Question: After using stolen code signing certificates to sign DUSTTRAP malware and components, is APT41 likely to use Windows services to execute DUSTPAN before using Windows Services with names such as Windows Defend for persistence of DUSTPAN? Answer: Yes

**[End of Examples]**

3. Provide the Question-Answer Pair:

- Please follow the output format:

"Question: <insert question here> Answer: <insert answer here>".

**[End of Steps]**

Following the steps above, please generate a question based on the CTI report and ATT&CK tactic, technique and procedures given below.

**Box 4: Few-shot prompt for question generation for *AttackSeq-Procedure-No*.**

You are a cybersecurity expert with deep knowledge of Cyber Threat Intelligence (CTI) reports and the MITRE ATT&CK framework.

**[Inputs]:**

You will receive two parts:

1. A Reference Question-Answer Pair that focuses on the logical sequence of TTPs in a CTI report, note that the answer to the this question is always "Yes".

2. A Reference MITRE TTP that is NOT supported in the CTI report.

**[End of Inputs]**

Your task is to generate two questions based on the given reference question, such that attack sequence described in the question is modified and the correct answer to the two questions is "No". The definitions of the two questions are as follows:

1. Question 1 should negate the "before" and/or "after" clauses of the reference question, such that the attack sequence contradicts the original sequence of TTPs in the reference question.

2. Question 2 should replace the main procedures in the reference question with the provided reference MITRE TTP, such that the replaced main procedures are not found in the CTI report.

**[Steps]:**

Please follow these steps:

1. Analyze the Reference Question-Answer Pair:

- Identify and outline the attack sequence in the order presented in the reference Question-Answer Pair.

2. Construct the Questions:

- Design two questions that modify the attack sequence described in the reference question.

- Ensure that the answer to the questions is "No".

- The question should be concise, clear, and targeted towards experienced cybersecurity professionals.

- Please refer to the examples below for guidance.

**[Examples]:**

Example Questions:

- Example 1:

Reference Question: After gaining initial access through compromised VPN accounts, will the Ke3chang malware most likely run commands on the command-line interface before achieving persistence by adding a Run key?

Reference Answer: Yes

Reference TTP: Tactic: Initial Access, Technique: T1651-Cloud Administration Command, Example Procedures: AADInternals can execute commands on Azure virtual machines using the VM agent. APT29 has used Azure Run Command and Azure Admin-on-Behalf-of (AOBO) to execute code on virtual machines. Pacu can run commands on EC2 instances using AWS Systems Manager Run Command.

Question 1: After achieving persistence by adding a Run key, will the Ke3chang malware run commands on the command-line interface only after gaining initial access through compromised VPN accounts? Answer: No

Question 2: After gaining initial access through compromised VPN accounts, will the Ke3chang malware most likely execute commands on Azure virtual machines using the VM agent before achieving persistence by adding a Run key? Answer: No

- Example 2:

Reference Question: Will Axiom acquire dynamic DNS services for use in the targeting of intended victims before gaining initial access to the victim's network using SQL injection?

Reference Answer: Yes

Reference TTP: Tactic: Resource Development, Technique: T1585.001-Social Media Accounts, Example Procedures: APT32 has set up Facebook pages in tandem with fake websites. Cleaver has created fake LinkedIn profiles that included profile photos, details, and connections. EXOTIC LILY has established social media profiles to mimic employees of targeted

companies.

Question 1: Will Axiom acquire dynamic DNS services for use in the targeting of intended victims only after gaining initial access to the victim's network using SQL injection? Reference Answer: No

Question 2: Will Axiom set up Facebook pages in tandem with fake websites before gaining initial access to the victim's network using SQL injection? Answer: No

- Example 3:

Reference Question: Will Ke3chang perform frequent and scheduled data exfiltration from compromised networks after establishing connection with the C2 server through Internet Explorer (IE) by using the COM interface IWebBrowser2?

Reference Answer: Yes

Reference TTP: Tactic: Exfiltration, Technique: T1030-Data Transfer Size Limits, Example Procedures: AppleSeed has divided files if the size is 0x1000000 bytes or more. APT28 has split archived exfiltration files into chunks smaller than 1MB. APT41 transfers post-exploitation files dividing the payload into fixed-size chunks to evade detection.

Question 1: Will Ke3chang perform frequent and scheduled data exfiltration from compromised networks only before establishing connection with the C2 server through Internet Explorer (IE) by using the COM interface IWebBrowser2? Answer: No

Question 2: Will Ke3chang divide files if the size is 0x1000000 bytes or more after establishing connection with the C2 server through Internet Explorer (IE) by using the COM interface IWebBrowser2? Answer: No

**[End of Examples]**

3. Provide the Question-Answer Pairs:

- Please follow the output format:

"Question 1: <insert question 1 here> Answer: <insert answer to question 1 here>."

"Question 2: <insert question 2 here> Answer: <insert answer to question 2 here>."

**[End of Steps]**

Following the steps above, please generate two questions based on the Reference Question-Answer Pair and Reference MITRE TTP given below. Please only provide the final output of the two questions.

## C.2 DATASET REFINEMENT PROMPT TEMPLATES

The prompt templates to filter based on the *Answerability* criteria is in Box 5, while the feedback and refinement prompts are in Box 6 and Box 7 respectively.

---

**Box 5: Prompt template for verifying *Answerability* during Self-Refinement.**

You are a cybersecurity expert with deep knowledge of Cyber Threat Intelligence (CTI) reports and the MITRE ATT&CK framework.

**[Inputs]**:

You will receive three parts:

1. CTI Outline: A structured account of a cyber attack, ordered by MITRE ATT&CK tactics. Additional context under "Others" provides background on the threat actor but is secondary.

2. TTP Description: A reference description of the correct answer corresponding to the question.

3. Question with Answer Choices: A question aimed at inferring a TTP from the attack sequence described in the CTI report, along with one correct answer and distractors among the answer choices.

**[End of Inputs]**

Your task is to evaluate the answerability of the given question using the provided information in the CTI Outline and TTP description. We define answerability based on three factors below:

1. The correct answer must be supported by the CTI outline.

2. The correct answer must clearly stand out as the best answer choice to the question based on the CTI outline.

3. Suppose the masked_tactic paragraph is removed from the CTI outline, the correct answer must be deducible from the answer choices by using the information provided in remaining tactics of the CTI outline and TTP description. You may also refer to your external cybersecurity knowledge to determine if the correct answer is deducible.

**[Steps]**:

Please follow these steps:

1. Analyze the CTI report:

- Read the report carefully.

- Identify and list the attack sequence in the order presented by the MITRE ATT&CK tactics.

2. Analyze the TTP Description:

- Read the TTP description of the correct answer carefully.

3. Evaluate the Question with Answer Choices:

- Read the question and the provided answer choices carefully.

- Match the correct answer with the provided TTP description.

- Determine step-by-step if the question is answerable based on the definition above. 3. Output evaluation result:

- Output one of the following:

- "A": Indicates that the question is answerable.

- "B": Indicates that the question is not answerable.

- "C": Indicates that you do not know/cannot determine if the question is answerable.

- Please also include a short and concise explanation of your evaluation result.

- Please follow the output format:

"Explanation: <insert explanation here> Evaluation Result: <insert letter here>."

**[End of Steps]**

Following the steps above, please evaluate the question using the CTI report and description below and only output the evaluation result.

---

**Box 6: Prompt template for assessing question quality based on the evaluation criteria.**

You are a cybersecurity expert with deep knowledge of Cyber Threat Intelligence (CTI) reports and the Tactics, Techniques and Procedures (TTPs) in MITRE ATT&CK framework.
**[Inputs]**:
You will receive three parts:
1. CTI Outline: A structured account of a cyber attack, ordered by MITRE ATT&CK tactics. Additional context under "Others" provides background on the threat actor but is secondary.
2. TTP Description: A reference description of the correct answer to the question.
3. Question with Answer Choices: A question aimed at inferring a TTP from the attack sequence described in the CTI outline, along with one correct answer and distractors among the answer choices.
**[End of Inputs]**

Your task is to evaluate the QA pair and provide your feedback for each of the criteria defined below:
**[Evaluation Criteria]**:
Please refer to the definition of each feedback criterion:
1. Clarity: Is the question precise, unambiguous, and free of vague phrasing? Does it avoid directly mentioning the correct answer, ensuring the respondent must infer the correct answer rather than having it stated in the question?
2. Logical: Does the question align with the logical sequence of MITRE ATT&CK tactics in the CTI outline? Does the question reference TTPs from the preceding or subsequent tactics in the CTI outline such that it logically leads to the correct answer?
3. Relevance: Does the question directly relate to the CTI outline?
4. Consistency: Does the question align with the provided TTP Description?
5. Answer Consistency: Can the question be fully answered using the correct answer, without any contradictions or inconsistencies?
**[End of Evaluation Criteria]**

**[Steps]**:
Please follow these steps:
1. Analyze the CTI outline:
- Read the CTI outline carefully.
- Identify and outline the attack sequence in the order presented by the MITRE ATT&CK tactics.
2. Analyze the TTP Description:
- Read the TTP description of the correct answer carefully.
3. Evaluate the Question with Answer Choices:
- Read the question and the provided answer choices carefully.
- Assess each criterion step by step, rating it on a scale of 1 to 5 (1 = poor, 5 = excellent).
- Provide a short and concise feedback for each rating.
4. Output Feedback Scores:
- Please follow the output format:
Feedback Scores:
- Clarity: <Your feedback> (<Score>/5)
- Logical: <Your feedback> (<Score>/5)
- Relevance: <Your feedback> (<Score>/5)
- Consistency: <Your feedback> (<Score>/5)
- Answer Consistency: <Your feedback> (<Score>/5)
Total Score: <Total Score>/25
**[End of Steps]**
Following the steps above, please evaluate the Question with Answer Choices below using the provided CTI report and TTP Description. Please only output the Feedback Scores.

**Box 7: Prompt template for question refinement.**

You are a cybersecurity expert with deep knowledge of Cyber Threat Intelligence (CTI) reports and the MITRE ATT&CK framework.

**[Inputs]:**

You will receive three parts:

1. CTI Outline: A structured account of a cyber attack, ordered by MITRE ATT&CK tactics. Additional context under "Others" provides background on the threat actor but is secondary.

2. Question with Answer Choices: A question aimed at inferring a TTP from the attack sequence described in the CTI report, along with one correct answer and distractors among the answer choices.

3. Feedback Results: A list of feedback scores and explanations for each desired criterion of the question defined below.

**[End of Inputs]**

Your task is to iteratively refine the quality of the given question based on the feedback provided in the Feedback Results.

**[Evaluation Criteria]:**

Please refer to the definition of each feedback criterion:

1. Clarity: Is the question precise, unambiguous, and free of vague phrasing? Does it avoid directly mentioning the correct answer, ensuring the respondent must infer the correct answer rather than having it stated in the question?

2. Logical: Does the question align with the logical sequence of MITRE ATT&CK tactics in the CTI outline? Does the question reference TTPs from the preceding or subsequent tactics in the CTI outline such that it logically leads to the correct answer?

3. Relevance: Does the question directly relate to the CTI outline?

4. Consistency: Does the question align with the provided TTP Description?

5. Answer Consistency: Can the question be fully answered using the correct answer, without any contradictions or inconsistencies?

**[End of Evaluation Criteria]**

**[Steps]:**

Please follow these steps:

1. Analyze the CTI report:
- Read the report carefully.

2. Analyze the Question with Answer Choices:
- Read the question and the provided answer choices carefully.

3. Analyze the Feedback Results:
- Based on the feedback given in each criterion, refine the question to improve the each aspect.
- Please ensure that the correct answer to the refined question is the same as the original question.
- Please also ensure that the question avoids hinting at the correct answer.

4. Output the Refined Question:
- Please follow the output format:
"Refined Question: <Your refined question here>."

**[End of Steps]**

Following the steps above, please refine the question based on the Feedback Results and CTI Outline provided below. Please only output the refined question.

## C.3 Automatic Evaluation Prompt Templates

We utilize the definitions in the evaluation criteria in Table 5 to create prompts. We show an example prompt template for evaluating the *Logical* aspect of the question shown in Box 8.

---

**Box 8: Prompt template for evaluating the *Logical* aspect.**

You are a cybersecurity expert with deep knowledge of Cyber Threat Intelligence (CTI) reports and the MITRE ATT&CK framework.
**[Inputs]:** You will receive three parts:
1. CTI Outline: A structured account of a cyber attack, ordered by MITRE ATT&CK tactics. Additional context under "Others" provides background on the threat actor but is secondary.
2. Question with Answer Choices: A question aimed at inferring a TTP from the attack sequence described in the CTI report, along with one correct answer and distractors among the answer choices.
3. Description to Correct Answer: The description of the correct answer from the MITRE ATT&CK framework.
**[End of Inputs]**

Your task is to rate the question on one metric below.
**[Definition]**
Evaluation Criteria:
Logical (1-5): Does the question align with the logical sequence of MITRE ATT&CK tactics in the CTI outline? Does the question reference TTPs from the preceding and/or subsequent tactics in the CTI outline such that it logically leads to the correct answer? The scale is defined as follows:
1 - Not Logical: The question does not align with the logical sequence of MITRE ATT&CK tactics in the CTI outline. It ignores or contradicts the natural order of tactics and TTPs.
2 - Weak Logical Alignment: The question shows minimal alignment with the MITRE ATT&CK sequence. It may reference unrelated tactics or disrupt the logical flow.
3 - Moderately Logical: The question has some logical alignment, but it may not reference preceding or subsequent tactics clearly. The sequence could be improved.
4 - Strong Logical Alignment: The question follows the expected sequence of MITRE ATT&CK tactics and references preceding or subsequent TTPs in a logical manner.
5 - Perfect Logical Alignment: The question perfectly aligns with the MITRE ATT&CK framework, referencing relevant TTPs in a way that naturally leads to the correct answer.
**[End of the Definition]**

**[Steps:]**
Evaluation Steps:
1. Analyze the CTI report and Description to Correct Answer:
- Read the report and the provided description carefully.
- Identify and list the attack sequence in the order presented by the MITRE ATT&CK tactics.
2. Evaluate the Question:
- Read the question and the provided answer choices carefully.
- Using the CTI outline and provided description to the correct answer, Rate the question on a scale of 1-5 according to the evaluation criteria above.
3. Output evaluation score:
- Please only output the numerical evaluation score based on the defined criteria.
**[End of Steps]**
Following the steps above, please evaluate the question and only output the numerical evaluation score.

---

## C.4 ANSWERING PROMPT TEMPLATES

The prompt templates for the three benchmark settings (*i.e., Context setting*, *Zero-Shot setting* and *RAG-empowered setting*) are shown in Box 9, Box 10, Box 11 respectively.

---

**Box 9: Prompt template for the *Context setting*.**

You are a cybersecurity expert with deep knowledge of Cyber Threat Intelligence (CTI) reports and the MITRE ATT&CK framework.
**[Inputs]:**
You will receive two parts:
1. A CTI Report that describe a cyber attack ordered by MITRE ATT&CK tactics. Note that additional information labeled as "Others" provides context about the threat actor but is secondary.
2. A Question about a sequence of TTPs with several answer choices.
**[End of Inputs]**

Your task is to determine which answer choice forms the most plausible sequence of TTPs based on the attack sequence described in the CTI report. Note that the CTI report contains key details required for your analysis, but it may not directly state the answer. Your evaluation of the answer choices is essential to arrive at the correct answer.
**[Steps:]**
Please follow these steps:
1. Analyze the CTI report:
- Read the report carefully.
- Identify and list the attack sequence in the order presented by the MITRE ATT&CK tactics.
2. Analyze the Question:
- Read the question and its answer choices.
- Identify the sequence of TTPs mentioned in the question.
3. Compare and Evaluate:
- Match the extracted attack sequence from the CTI report with the details in the question.
- Evaluate each answer choice to determine which one aligns best with the attack sequence and any critical contextual information.
4. Provide a Step-by-Step Reasoning and Final Answer:
- Outline your reasoning step-by-step.
- Conclude with the final answer in the following format:
"Final Answer: <insert answer choice here>."
**[End of Steps]**
Following the steps above, please answer the question below using the provided CTI report.

---

---

**Box 10: Prompt template for the *Zero-Shot setting*.**

You are a cybersecurity expert with deep knowledge of Cyber Threat Intelligence (CTI) reports and the MITRE ATT&CK framework.
**[Inputs]:**
You will receive a question about a sequence of TTPs with several answer choices.
**[End of Inputs]**

Your task is to determine which answer choice forms the most plausible sequence of TTPs based on the attack sequence described in the question.
**[Steps:]**
Please follow these steps:
1. Analyze the Question:
- Read the question and its answer choices.
- Identify the sequence of TTPs mentioned in the question.
2. Compare and Evaluate:
- Evaluate each answer choice to determine which one aligns best with the attack sequence in the question.
3. Provide a Step-by-Step Reasoning and Final Answer:
- Outline your reasoning step-by-step.
- Conclude with the final answer in the following format:
"Final Answer: <insert answer choice here>."
**[End of Steps]**
Following the steps above, please answer the question below.

---

**Box 11: Prompt template for the *RAG-empowered setting*.**

You are a cybersecurity expert with deep knowledge of Cyber Threat Intelligence (CTI) reports and the MITRE ATT&CK framework.
**[Inputs]:**
You will receive two parts:
1. A Question about a sequence of TTPs with several answer choices.
2. A list of Related TTPs that are relevant to the question.
**[End of Inputs]**

Your task is to determine which answer choice forms the most plausible sequence of TTPs based on the attack sequence described in the question.
**[Steps:]**
Please follow these steps:
1. Analyze the Question:
- Carefully read the question and its answer choices.
2. Analyze the Related TTPs:
- Analyze the list of Related TTPs to understand the context of the question.
3. Compare and Evaluate:
- Based on the related TTPs, evaluate each answer choice to determine which one aligns best with the attack sequence in the question.
4. Provide a Step-by-Step Reasoning and Final Answer:
- Outline your reasoning step-by-step.
- Conclude with the final answer in the following format:
"Final Answer: <insert answer choice here>."
**[End of Steps]**
Following the steps above, please answer the question below.

