# OpenReview forum: "AttackSeqBench: Benchmarking Large Language Models in Analyzing Attack Sequences within Cyber Threat Intelligence"
_ICLR.cc/2026/Conference — ICLR 2026 Conference Withdrawn Submission_

### Official Review · Reviewer_rLu1 · 2025-10-27

**Soundness:** 2
**Presentation:** 2
**Contribution:** 2
**Rating:** 2
**Confidence:** 5

**Summary:**

This paper introduces AttackSeqBench, a QA benchmark for cyber threat kill chain investigation built upon publicly available CTI reports. The authors collected multiple CTI reports and use LLM to extract the corresponding attack Tactic/Technique/Procedure behind the natural language sequences. They then use LLM to automatically generate single-hop questions and answers based on the extracted information with evaluation and refinement. Finally, the paper evaluates several LLMs and strategies on AttackSeqBench to analyze their performance.

**Strengths:**

Provide a benchmark that focuses on cyber threat kill chain investigation, which is an important area in cybersecurity and not seen before in LLM benchmarks.

**Weaknesses:**

1. The benchmark relies on GPT-4o to generate Q&A items and also to include GPT-4o as one of the evaluated models. This creates a risk of model overlap between data construction and evaluation.

2. The pipeline converts CTI narratives into ATT&CK Tactics/Techniques/Procedures via an LLM-based KG/extraction framework, but the paper does not report precision/recall of this mapping. If a non-trivial fraction of extracted TTPs is wrong, downstream QA quality is directly affected.

3. For Procedure-No questions, even human experts only achieve an accuracy of 0.56, which is not much better than random guessing at 0.5. This suggests that there may be inherent issues with the questions themselves.

4. The utility of the detected attack sequences is questionable. It is not clear about the utility of the attack sequence extracted from the reports. In real attacks, it is extremely difficult for an attack with multiple steps to exactly match each step of an attack in a CTI report. In fact, separating an attack into multiple stages/techniques improves the probability of matching a reported technique to a technique observed in a real attack. This is also the main reason for MITRE to publish the matrix where each stage can have multiple techniques and are not limited to specific combinations of a few techniques.

5. The paper lacks demonstration in practical applications. How the attack sequences can be used is not explained in the paper. Obviously, we cannot hope to find an attack in the future that exactly matches the steps found in this paper’s detected attack sequences. Then how effectively can the attack sequences help attack investigation is not clear. For example, various types of logs, such as command line logs and network logs, record traces of actual attack steps. And recently, system audit logs that collect system call information can further connect these attack steps. However, these logs cannot be easily matched to the detected attack sequences since there are semantic gaps between reported attack steps and the system activities recorded in the logs (e.g., process names, file paths, IP addresses). The authors should explore deeper on these aspects to improve their benchmark.

**Questions:**

1. Would using a reasoning model for question generation improve item quality?

2. How do you explain the low human accuracy on Procedure-No questions?

3. Can you explain the motivation of the attack sequences from CTI reports?

4. How can these sequences be used to assist real-world attack investigation?

---

### Official Review · Reviewer_vnNQ · 2025-10-29

**Soundness:** 2
**Presentation:** 2
**Contribution:** 2
**Rating:** 2
**Confidence:** 4

**Summary:**

The paper introduces AttackSeqBench, a  benchmark for evaluating LLMs on their ability to understand and reason over adversarial behavior sequences described in real-world CTI reports. The benchmark focuses on three reasoning dimensions: tactical, technical, and procedural, and defines corresponding question-answering tasks aligned with the MITRE ATT&CK framework. Using  408 CTI reports, the authors develop an automated Q&A generation and refinement pipeline and evaluate seven LLMs, five large reasoning models (LRMs), and multiple post-training strategies across zero-shot, context-augmented, and retrieval-augmented (RAG) settings. Results show that current models, including reasoning-optimized ones, struggle to generalize across multi-stage attack reasoning.

**Strengths:**

The paper formulates attack sequence analysis as a structured reasoning benchmark for LLMs, extending prior CTI benchmarks beyond simple entity extraction or classification tasks. It introduces a scalable framework for constructing TTP-based benchmarks from public CTI reports through automated Q&A generation, refinement, and evaluation aligned with the MITRE ATT&CK framework. The evaluation comprehensively compares diverse LLMs, large reasoning models, and post-training strategies across zero-shot, contextual, and retrieval-augmented settings.

**Weaknesses:**

- **Overreliance on LLM-based dataset construction:** The benchmark depends heavily on LLMs for both attack sequence construction and question generation, yet the paper does not adequately discuss the inherent limitations of this approach. Since the initial attack sequence extraction step is critical to dataset validity, relying on models with limited extraction accuracy raises concerns about noise propagation. For instance, the referenced AttacKG+ framework reports an F1-score of only 56.6%, suggesting that LLM-based extraction can introduce significant factual or structural inconsistencies that may affect benchmark quality.

- **Task formulation limitations:** Although the benchmark aims to evaluate attack sequence reasoning, the task structure, multiple-choice, and yes/no Q&A primarily test factual recall and event sequencing rather than genuine multi-hop or causal reasoning. Moreover, since the benchmark is based on historical incidents, certain tasks (e.g., identifying techniques between two observed events) can be inherently ambiguous, making it unclear how well this setup generalizes to predictive or forward-looking CTI reasoning, which would be of greater practical relevance.

- **Limited analysis of failure modes:** The evaluation focuses largely on accuracy metrics without providing a deeper qualitative breakdown of model errors, such as confusion between tactics, reasoning gaps, or noise in retrieved context. A more detailed failure taxonomy or per-tactic performance analysis could offer insights into specific reasoning weaknesses and guide targeted model improvements.

- **Shallow treatment of RAG and post-training strategies:** While the paper observes that retrieval and fine-tuning underperform, it does not analyze the root causes behind these results. Factors such as embedding misalignment, retrieval noise, or insufficient context integration may explain this behavior. Incorporating ablation studies, retrieval relevance evaluations, or context-quality diagnostics would make the findings more informative and actionable.

**Questions:**

- For the RLVR experiments, how were the task rewards designed?

---

### Official Review · Reviewer_kRQn · 2025-10-31

**Soundness:** 2
**Presentation:** 1
**Contribution:** 4
**Rating:** 4
**Confidence:** 4

**Summary:**

This paper presents *AttackSeqBench*, a Q&A benchmarks to evaluate AI capabilities on *attack sequence* question answering. This work is relevant as AI is increasingly used in cyber-threat intelligence (CTI) to make sense of dense and unstructured CTI reports. The paper performs a comprehensive evaluation of LLMs and LRMs on the task and highlights knowledge gaps.

**Strengths:**

- The paper provides a comprehensive evaluation of various kinds of LLMs and LRMs, including hyperparameter tuning experiments to reveal significant gaps in CTI capabilities
- Human cross-evaluation of the Q&A benchmark with high Likert scores strengthens the claim that the benchmark is  viable for evaluating CTI capabilities
- Introduction of the AttackSeq-Procedure-No questions highlights interesting gaps in LLM reasoning capabilities where the performance drops on these negated questions

**Weaknesses:**

- Comparing human scores with LLM and LRM zero-shot scores reveals that the models already outperform on tactics, techniques, procedures-yes, and procedures-no. This indicates that the LLM generated questions are better suited for LLM answering rather than human answering, which may indicate an undue bias in the benchmark and break its viability
- The paper should consider the evolution of MITRE ATT&CK KB across different versions which may affect the Zero-shot and Context-aware settings as different models may have a different picture of the MITRE ATT&CK KB depending on their knowledge cutoff date. For context, MITRE ATT&CK has gone from version 14 in 2023 to version 18 currently, where certain TTP numbers may have been added, changed, or deleted.
- There are substantial flaws in presentation of the paper contents that require frequently switching back and forth between the main text and the appendices to correctly understand the approach. For instance, Figure 1 and corresponding acronyms like KB, KG are not explained in the main text, Table 1 Likert scores cannot be deciphered without appendix A.2. There are  more such instances.

**Questions:**

- Figure 1 is not explained adequately in the text, for instance why are there 3 victims? What are the purpose of staging server if there is a C2 server?
- Line 107: KB is not defined, is it knowledge base?
- Figure 2, Line 138: KG is not defined, is it knowledge graph?
- Some typos and grammar mistakes such as "combin", "even severer", "pf"
- Line 179 mentions 3 categories however there should be as many as the columns in Table 1. Also the text should clearly list the abbreviations as they are not obvious
- According to the AttacKG+ paper, they claim to process 500 CTI reports, so the claim (line 1125) that AttackSeqBench processes substantially more CTI reports is false

---

### Official Review · Reviewer_ipWo · 2025-11-01

**Soundness:** 2
**Presentation:** 3
**Contribution:** 2
**Rating:** 2
**Confidence:** 5

**Summary:**

This paper introduces a benchmark for evaluating LLMs' ability to analyze cyber attack sequences. The benchmark evaluates models on TTPs using question-answering tasks across multiple settings (zero-shot, context-based, and retrieval-augmented). The authors test 7 LLMs, 5 LRMs, and 4 post-training strategies. The results help highlight how current models struggle with reasoning over sequential attack behaviors.

**Strengths:**

1. The paper tests multiple models and settings (zero-shot, context, RAG), providing a deep dive into current model limitations.

2. The paper includes an automated pipeline for QA construction and refinement, making the benchmark extensible.

**Weaknesses:**

1. The benchmark focuses only on QA tasks, which may oversimplify real-world CTI workflows. In practice, CTI analysts need summarization, timeline reconstruction, and prediction, thus not only question answering.

2. The work is heavily tied to the ATT&CK framework, which is not the only threat modeling method (e.g., Diamond Model, Lockheed Martin Kill Chain). This may limit adaptability to other schemas.

3. The study lacks validation with actual security analysts to show whether improvements on the benchmark lead to meaningful utility in real-world workflows.

4. The automated QA generation and refinement pipeline uses LLMs. This introduces unknown errors, biases, or synthetic evidence into the benchmark dataset itself.

5. The paper shows RAG performs poorly, but offers limited discussions or insights on how to improve retrieval-augmented reasoning in this domain.

**Questions:**

Please see the comments above.

---

### Note · Authors · 2025-12-01

I have read and agree with the venue's withdrawal policy on behalf of myself and my co-authors.